# FEDERATED CONTINUAL NOVEL CLASS LEARNING

## ABSTRACT

In a privacy-focused era, Federated Learning (FL) has emerged as a promising machine learning technique. However, most existing FL studies assume that the data distribution remains nearly fixed over time, while real-world scenarios often involve dynamic and continual changes. To equip FL systems with continual model evolution capabilities, enabling them to discover and incorporate unseen novel classes, we focus on an important problem called *Federated Novel Class Learning* (FedNovel) in this work. The biggest challenge in FedNovel is to merge and align novel classes that are discovered and learned by different clients without compromising privacy. To address this, we propose a *Global Alignment Learning* (GAL) framework that can accurately estimate the global novel class number and provide effective guidance for local training from a global perspective, all while maintaining privacy protection. Specifically, GAL first locates high-density regions in the representation space through a bi-level clustering mechanism to estimate the novel class number, with which the global prototypes corresponding to novel classes can be constructed. Then, GAL uses a novel semantic weighted loss to capture all possible correlations between these prototypes and the training data for mitigating the impact of pseudo-label noise and data heterogeneity. Extensive experiments on various datasets demonstrate GAL's superior performance over state-of-the-art novel class discovery methods. In particular, GAL achieves significant improvements in novel-class performance, increasing the accuracy by 5.1% to 10.6% in the case of one novel class learning stage and by 7.8% to 17.9% in the case of two novel class learning stages, without sacrificing known-class performance. Moreover, GAL is shown to be effective in equipping a variety of different mainstream FL algorithms with novel class discovery and learning capability, highlighting its potential for many real-world applications.

## 1 INTRODUCTION

Federated learning (FL) (McMahan et al., 2017) has gained significant popularity in recent years as a privacy-preserving machine learning (ML) technique. By enabling distributed devices or nodes to collaboratively train models without sharing raw data, FL offers a powerful solution to the growing need for data protection in a wide range of applications. Nearly all existing FL studies (Ghosh et al., 2020; Chen et al., 2021; Huang et al., 2022; Li et al., 2021b) assume that the system data distribution is largely fixed and known or can be easily estimated. However, in many real-world scenarios, the data distribution is intrinsically dynamic as new information becomes available, which poses unique challenges that need to be addressed to fully harness the potential of FL (Yoon et al., 2021; Qi et al.; Dong et al., 2023). In particular, this dynamic nature of data calls for the capability to discover the demands of new functionalities and then incorporate them in FL, enabling the system to adapt and stay relevant in the face of emerging trends and patterns. The importance of this capability becomes even more apparent when considering critical application domains such as healthcare, finance, transportation, and the Internet of Things. For instance, in the healthcare domain, disease diagnosis models built using FL must be able to cope with the continual emergence of novel diseases or rare conditions (Xu et al., 2021; Rieke et al., 2020), and an FL system capable of discovering and learning novel diseases could have greatly aided in the early detection and containment of pandemic outbreaks. A well-evolving FL system could also help healthcare professionals make more accurate diagnoses and ultimately provide better patient care.

To address the above issue and enable FL to cope with the continually-emerging novel functionality demand, we study an important problem called *Federated Novel Class Learning* (FedNovel) in

this work. `FedNovel` considers the following scenario. An FL system initially consists of multiple participants that conduct supervised training on their local data for a known label space. The data distributions of participants are non-independent and non-identical (non-IID) (McMahan et al., 2017). After sufficient training, the FL model is expected to perform well on known class data. At this moment, the FL administrator often stops the training to save time, labor, and other resources, and deploys the latest global model to serve all participants (McMahan et al., 2017). During the model usage, users feed unlabeled data into the model and receive inferred results. However, in real-world deployment scenarios of FL as mentioned, in addition to known class data, the participants often receive data belonging to unseen novel classes that also conform to a non-IID setting over time. As expected, ***such novel class data will be assigned with incorrect known labels. Thus, `FedNovel` aims to develop methods that can guide the FL framework to discover and learn these novel class data***, ultimately enabling the model to cluster novel data into different groups. At the same time, `FedNovel` also tries to preserve the model's good performance on known classes.

A seemingly straightforward approach for addressing `FedNovel` is to integrate FL and standard novel class discovery and learning (NCDL) (Li et al., 2023; Roy et al., 2022; Zhang et al., 2022). However, this solution may not be feasible due to multiple reasons. First, in order to cluster novel data, it is necessary to know the number of novel classes. Conventional NCDL typically assumes that such information is known (Li et al., 2023; Sun & Li, 2022; Roy et al., 2022), as the model trainer can access the training data and is aware of the task that the model will be applied to. However, this assumption is unreasonable in FL, as the FL administrator cannot access the data and participants lack the expertise to estimate task details (McMahan et al., 2017; Dong et al., 2022). Second, due to the non-IID setting, even if participants can perform NCDL locally, the novel classes discovered by each participant are likely different – meaning that there could be varying degrees of overlap that makes merging their novel label spaces challenging. Furthermore, we discovered that when applying existing NCDL methods, the non-IID setting increases the distances between representations of the same novel class held by different participants, instead of bringing them closer as desired. This further exacerbates the difficulty of merging local novel label spaces.

To overcome these challenges, we design a novel framework called *Global Alignment Learning* (`GAL`), which can accurately estimate the novel class number and effectively guide local training at a global level to tackle the non-IID heterogeneity while preserving privacy. The overview of `GAL` is shown in Figure 1. Specifically, `GAL` first needs participants to find potential cluster centroids in their local data using unsupervised clustering. These local centroids are synthetic virtual points that do not correspond to real data and need to be sent to the server only once. We then design an approach called *Potential Prototype Merge* (`PPM`) to find high-density regions in all local centroids received by the server, obtaining an estimated novel class number on the global side. With this estimation, we initialize the corresponding global novel class prototypes and propose a novel *Semantic-Weighted Loss* (`SWL`) that can guide the local training to effectively align with these global prototypes, thereby alleviating the impact of non-IID. We conduct extensive experiments on benchmark datasets to validate `GAL`'s performance. In comparison with a number of state-of-the-art NCDL methods, `GAL` provides significant improvements on novel class accuracy in solving `FedNovel`, with the absolute improvements ranging from 5.1% to 10.6% for the case of one novel class learning stage and 7.8%-17.9% for the case of two novel class learning stages, while nearly without forgetting on known classes. Moreover, `FedNovel` can equip a variety of mainstream FL algorithms with strong NCDL capability in our experiments. In summary, our contributions include:

- We address an important problem in FL called *Fed**erated* *Novel* *Class* *L**earning* (`FedNovel`), in which the biggest challenge is to merge and align novel classes of different participants that conform to a non-IID setting without compromising privacy.

- We design a *Global Alignment Learning* (`GAL`) framework to tackle the `FedNovel` problem. `GAL` mainly consists of a novel class number estimation approach *Potential Prototype Merge* that can find high-density regions in potential prototypes, and a *Semantic-Weighted Loss* that considers the semantic relationship between the data and all novel classes to align local training.

- We validate `GAL` on multiple datasets and various `FedNovel` scenarios. `GAL` achieves significant performance improvements over a number of state-of-the-art novel class learning baselines. Moreover, besides the standard FedAvg (McMahan et al., 2017) algorithm, `GAL` is highly effective in equipping a number of other FL algorithms with the capability of novel class learning.

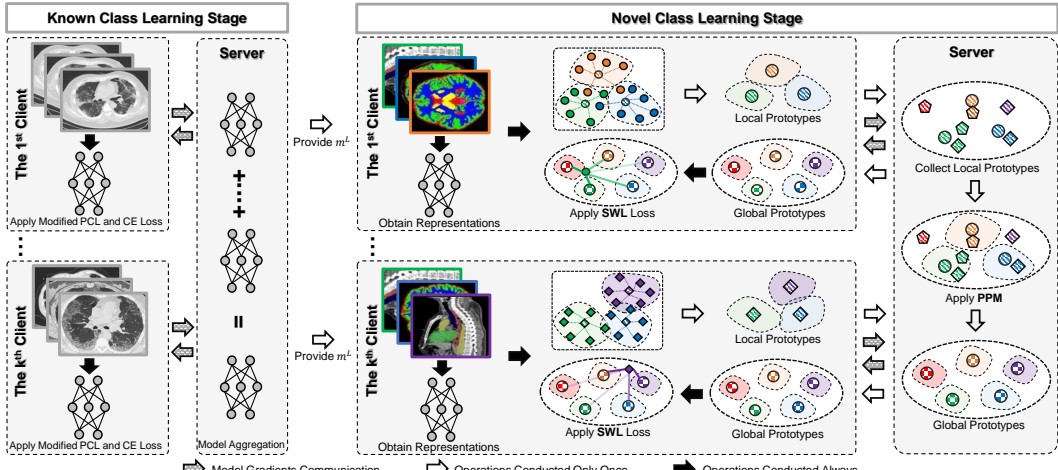

Figure 1: Overview of GAL for enabling FL models to discover and learn novel classes. When the known class learning finishes and the novel class learning starts, FL participants first apply unsupervised clustering on their local data to find local potential novel prototypes. These local prototypes are sent to the server only once and GAL applies PPM to merge these prototypes into global novel prototypes. After initializing the classifier with global prototypes, each selected client leverages SWL to conduct local training.

## 2 RELATED WORK

**Federated Learning** (FL) research (Ghosh et al., 2020; Huang et al., 2022; Li et al., 2021b) is primarily concentrated on static scenarios where the FL system is assumed to not undergo significant changes over time. This assumption is overly idealistic and not reflective of many real-world scenarios. Although there are studies (Nguyen et al., 2022; Jiang & Lin, 2022; Yu et al., 2023) that employ adversarial training to enhance the out-of-distribution generalization of FL models, and Federated Openworld / Openset Semi-Supervised Learning (Zhang et al., 2023; Pu et al., 2023) tries to enable the learning of unseen data categories. However, these studies typically only work in supervised static FL scenarios, while existing unsupervised FL approaches (Zhang et al., 2020; Lubana et al., 2022) only consider static situations and primarily focus on training representation models, with downstream tasks requiring further training. Recently, there has been a surge in research exploring dynamic FL (Dong et al., 2022; Yoon et al., 2021; Qi et al.), where the data distribution of FL systems is subject to dynamic changes over time, but these studies still focus on the supervised setting. In summary, current FL works lack the ability to discover and learn novel class data in unsupervised scenarios, and prevent significant forgetting on learned knowledge. In contrast, our proposed GAL framework effectively drives FL models to discover and learn novel classes, thereby adapting them to the dynamic changes in the real world.

**Novel Class Discovery and Learning** (NCDL) refers to the process through which ML models identify and learn previously unseen data classes (Han et al., 2021). Existing NCDL studies can generally be divided into label-available and label-unavailable cases. Label-available NCDL is also known as Openworld Semi-Supervised Learning (SSL) (Cao et al., 2022; Guo et al., 2022), which typically mixes labeled known class and unlabeled novel class data together, and employs SSL (Yang et al., 2022; Li et al., 2023), clustering (Vaze et al., 2022), or weakly-supervised learning (An et al., 2022; Chi et al.) to facilitate novel class learning. In contrast, the less-explored label-unavailable NCDL focuses on practical scenarios where labeled known class data become inaccessible due to privacy protection or cost reduction (Rebuffi et al., 2017). In these works, models usually undergo two stages: first supervised learning with labeled known class data and then pseudo-label or self-supervised learning (Zhang et al., 2022; Roy et al., 2022; Zhao & Mac Aodha, 2023) on unlabeled novel class data. FedNovel is a label-unavailable NCDL problem, but data heterogeneity in FL heavily reduces the effectiveness of existing NCDL methods. Thus, to tackle the unique challenges in FedNovel, we propose the GAL framework that can effectively guide and align the local training of different FL participants, enabling more effective novel class learning.

## 3 METHODOLOGY

### 3.1 PRELIMINARIES

**Novel Class Discovery and Learning (NCDL).** In NCDL, suppose there is a labeled dataset $\mathcal{D}^{\mathrm{L}} = \{(\boldsymbol{x}_i^{\mathrm{L}}, \boldsymbol{y}_i^{\mathrm{L}})\}_{i=1}^{N^{\mathrm{L}}}$, and all labels belong to a one-hot label space of dimension $C^{\mathrm{L}}$, i.e., $\boldsymbol{y}_i^{\mathrm{L}} \in \mathcal{Y}^{\mathrm{L}}$. Moreover, the input marginal distribution of $\mathcal{D}^{\mathrm{L}}$ is $\mathcal{X}^{\mathrm{L}}$. Besides, there is another unlabeled dataset $\mathcal{D}^{\mathrm{U}} = \{\boldsymbol{x}_i^{\mathrm{U}}\}_{i=1}^{N^{\mathrm{U}}}$, which has $C^{\mathrm{U}}$ unique classes in $\mathcal{Y}^{\mathrm{U}}$, and its input distribution is $\mathcal{X}^{\mathrm{U}}$. As in most NCDL studies, it is assumed that the labels in $\mathcal{Y}^{\mathrm{L}}$ and $\mathcal{Y}^{\mathrm{U}}$ are disjoint, i.e., $\mathcal{Y}^{\mathrm{L}} \cap \mathcal{Y}^{\mathrm{U}} = \emptyset$. The labels in $\mathcal{Y}^{\mathrm{L}}$ and $\mathcal{Y}^{\mathrm{U}}$ are usually called known classes and novel classes, respectively. The general objective of NCDL is to partition the data belonging to novel classes into different clusters with the help of $\mathcal{D}^{\mathrm{L}}$. More specifically, we leverage a neural network $m = f_\theta \circ g_\omega$, which consists of a feature extractor $f$ parameterized on $\theta$ and a classifier $g$ parameterized on $\omega$, to achieve NCDL:

$$\max_{\theta,\omega} \mathbb{E}_{\boldsymbol{x}_i, \boldsymbol{x}_j \sim \mathcal{X}^{\mathrm{U}}; \boldsymbol{y}_i, \boldsymbol{y}_j \sim \mathcal{Y}^{\mathrm{U}}} \{\mathbb{I}[g_\omega(f_\theta(\boldsymbol{x}_i)) = g_\omega(f_\theta(\boldsymbol{x}_j))] \cdot \mathbb{I}[\boldsymbol{y}_i = \boldsymbol{y}_j]\}, \tag{1}$$

where $\mathbb{I}[\cdot]$ is an indicator function that if the inner condition is true, $\mathbb{I}[\mathrm{True}] = 1$, otherwise $\mathbb{I}[\mathrm{False}] = 0$. $\boldsymbol{y}_i, \boldsymbol{y}_j$ are the ground truth labels of novel samples $\boldsymbol{x}_i, \boldsymbol{x}_j$, though unavailable during NCDL.

**Federated Learning (FL).** At the beginning of FL, we assume that a global model is trained across $K$ participants, and each participant holds its own local dataset $\mathcal{D}^k = \{(\boldsymbol{x}_i^k, \boldsymbol{y}_i^k)\}_{i=1}^{N^k}$. The data distributions of participants are non-independent and non-identically distributed (i.e., non-IID) from each other (McMahan et al., 2017; Hsu et al., 2019). In this case, the label spaces of different participants $\mathcal{Y}^k$ are not the same, but they all belong to a unified space $\mathcal{Y}^{\mathrm{L}}$, i.e., $\mathcal{Y}^k \subseteq \mathcal{Y}^{\mathrm{L}}$, which can be regarded as known classes in NCDL. For every global round, a random set of participants is selected as clients, namely $\mathcal{K}$, and they are required to conduct local training and then upload the model updates to the server. We assume that the local training is conducted with the CrossEntropy loss and the server applies FedAvg (McMahan et al., 2017) to aggregate model updates as follows:

$$\mathcal{L}_{\mathrm{CE}}^k = -\frac{1}{N_B^k} \sum_{i=1}^{N_B^k} \boldsymbol{y}_i^k \log g_\omega(f_\theta(\boldsymbol{x}_i^k)), \qquad \nabla_{\theta,\omega}^{\mathrm{Avg}} \mathcal{L}_{\mathrm{CE}} = \frac{1}{|\mathcal{K}|} \sum_{k \in \mathcal{K}} \nabla_{\theta,\omega} \mathcal{L}_{\mathrm{CE}}^k, \tag{2}$$

where $N_B^k$ is the sample number of a mini-batch for client $k$.

### 3.2 PROBLEM FORMULATION OF FedNovel

After training with Eq. 2 in a sufficient number of rounds, which we call *Known Class Learning* $\mathcal{T}^{\mathrm{L}}$, the FL model $m$ is hopefully able to perform well on $\mathcal{Y}^{\mathrm{L}}$. Then, the FL administrator may stop the training to save resources and deploy the latest model $m^{\mathrm{L}}$ to serve all participants in the FL framework. During the usage of $m^{\mathrm{L}}$, participants feed it with unlabeled testing data $\boldsymbol{x}^{\mathrm{test}}$ and obtain corresponding inference results $g_\omega(f_\theta(\boldsymbol{x}^{\mathrm{test}}))$. Intuitively, if $\boldsymbol{x}^{\mathrm{test}}$ belongs to the known classes ($\mathcal{Y}^{\mathrm{L}}$), the inference results should be mostly correct and accurate, i.e., $\mathbb{I}[g_\omega(f_\theta(\boldsymbol{x}^{\mathrm{test}})) = \boldsymbol{y}^{\mathrm{test}}] = 1$. However, if $\boldsymbol{x}^{\mathrm{test}}$ comes from unseen novel classes, the current model $m^{\mathrm{L}}$ will classify it into a certain known class, in other words, 100% misclassification ($\mathbb{I}[g_\omega(f_\theta(\boldsymbol{x}^{\mathrm{test}})) = \boldsymbol{y}^{\mathrm{test}}] = 0$).

Now let us more formally define the problem of FedNovel. Suppose that the FL training periodically steps into a stage called *Novel Class Learning* $\mathcal{T}^{\mathrm{U}}$, and we assume that there are $K^{\mathrm{U}}$ participants available in total in $\mathcal{T}^{\mathrm{U}}$. Note that $K^{\mathrm{U}}$ is uncertain and may be changing dynamically, especially since old participants are free to withdraw and new participants are welcome to join in at any time (Dong et al., 2022). At this moment, we denote the local novel class dataset of participant $k$ as $\mathcal{D}^{\mathrm{U},k} = \{\boldsymbol{x}_i^{\mathrm{U},k}\}_{i=1}^{N^{\mathrm{U},k}}$, screened from its received unlabeled testing dataset $\mathcal{D}^{\mathrm{test}}$. Consistent with the non-IID setting, although $\mathcal{D}^{\mathrm{U},k}$ has no labels, its ground-truth label space $\mathcal{Y}^{\mathrm{U},k}$ is distinct across different participants but also belongs to a unified novel label space $\mathcal{Y}^{\mathrm{U}}$. Our objective is to conduct effective NCDL that can cluster novel class data into different classes and preserve good performance on known classes as well:

$$\max_{\theta,\omega} \left\{ \sum_{\boldsymbol{x}_i, \boldsymbol{x}_j \in \mathcal{D}^{\mathrm{U}}} \mathbb{I}[g_\omega(f_\theta(\boldsymbol{x}_i)) = g_\omega(f_\theta(\boldsymbol{x}_j))] \cdot \mathbb{I}[\boldsymbol{y}_i = \boldsymbol{y}_j] + \mathbb{E}_{\boldsymbol{x} \sim \mathcal{X}^{\mathrm{L}}, \boldsymbol{y} \sim \mathcal{Y}^{\mathrm{L}}} \mathbb{I}[g_\omega(f_\theta(\boldsymbol{x})) = \boldsymbol{y}] \right\}. \tag{3}$$

### 3.3 OUR PROPOSED GAL FRAMEWORK

#### 3.3.1 REPRESENTATION ENHANCEMENT FROM KNOWN CLASS LEARNING

In order to achieve better NCDL, model $m^{\mathrm{L}}$ should not only have a good performance on known classes but also extract high-quality representations for novel classes. To this end, we modify Prototype Contrastive Learning (PCL) (Li et al., 2020a) to enhance the training in $\mathcal{T}^{\mathrm{L}}$. More specifically, our modified PCL constructs positive pairs as a sample representation $z_i = f_\theta(x_i)$ and its corresponding class prototype $p_{y_i}^{\mathrm{L}}$. The negative pairs consist of two parts: the first part includes all pairs between $z_i$ and all class prototypes $\{p_c^{\mathrm{L}}\}_{c=1}^{C^{\mathrm{L}}}$, and the second part includes sample-wise pairs between $z_i$ and samples belonging to other labels, which is shown as follows:

$$\mathcal{L}_{\mathrm{PCL}}^{\mathrm{L},k} = -\frac{1}{N_B^{\mathrm{L},k}} \sum_{i=1}^{N_B^{\mathrm{L},k}} \frac{\exp(z_i^{\mathrm{L},k} \cdot p_{y_i^{\mathrm{L},k}}^{\mathrm{L},k}/\tau)}{\sum_{c=1}^{C^{\mathrm{L}}} \exp(z_i^{\mathrm{L},k} \cdot p_c^{\mathrm{L}}/\tau) + \sum_{j=1}^{N_i^-} \exp(z_i \cdot z_j/\tau)}, \tag{4}$$

where $\tau = 0.07$ (consistent with Li et al. (2020a)) is a temperature parameter and $N_i^-$ is the set size of samples with different labels from $x_i$. Then we apply a combined loss $\mathcal{L}_{\mathcal{T}^{\mathrm{L}}} = \mathcal{L}_{\mathrm{CE}} + \eta \mathcal{L}_{\mathrm{PCL}}$ to train $m$ on known class data ($\eta = 0.1$; please refer to the appendix for sensitivity analysis).

#### 3.3.2 POTENTIAL PROTOTYPE MERGE

**Novel Class Data Filtering.** When stepping into *Novel Class Learning* $\mathcal{T}^{\mathrm{U}}$, it is possible for every participant to continually receive novel class and known class data. Thus we first need to separate known and novel classes from $\mathcal{D}^{\mathrm{test},k}$. Specifically, for each participant $k$, we assume that there is a data memory $\mathcal{M}^{k1}$ employed to store $\mathcal{D}^{\mathrm{U},k}$ that is filtered out during model $m^{\mathrm{L}}$ usage. We adopt a popular strategy (Zhang et al., 2022) based on the representation distance between data samples and known class prototypes. The traditional class prototype (Zhang et al., 2022; Roy et al., 2022) is constructed by averaging class-wise representations, but such construction is infeasible in FL as the data is distributed among participants. We solve this problem by viewing the first $C^{\mathrm{L}}$ dimensions of neuron weights $\omega$ of classifier $g$ as the prototypes $\mathcal{P}^{\mathrm{L}} = \{p_c^{\mathrm{L}}\}_{c=1}^{C^{\mathrm{L}}}$, if $g$ is composed of only one single linear layer without bias (Yao et al., 2022). The detailed filtering mechanism is as follows:

$$\mathcal{D}^{\mathrm{U},k} = \left\{ x \in \mathcal{D}^{\mathrm{test},k} \,\middle|\, \max \left\{ \frac{f_\theta(x) \cdot p_c^{\mathrm{L}}}{\|f_\theta(x)\|\|p_c^{\mathrm{L}}\|} \right\}_{c=1}^{C^{\mathrm{L}}} < r \right\}, \tag{5}$$

where $r = 0.5$ is a threshold. This implies that if a sample $x$ has the largest similarity with known prototypes lower than $r$, it should belong to novel classes. As for the screened known class data samples, there is no dedicated design, but we test GAL with applying Softmax-based pseudo labeling on them during $\mathcal{T}^{\mathrm{U}}$ (detailed results can be found in the ablation study settings of the appendix).

**Challenges of Novel Class Number Estimation in FedNovel.** As previously mentioned, the goal of NCDL is to accurately classify novel data into different clusters, which requires prior knowledge of the number of clusters. Most existing NCDL studies (Roy et al., 2022; Li et al., 2023; Yang et al., 2022) directly assume that such information is known, as regular NCDL often assumes that data collectors can access the training data and often possess sufficient expertise in the tasks they train. However, this assumption is unreasonable in FedNovel, and the novel class number estimation needs to tackle the following challenges:

- FL system administrator cannot access any training data and cannot predict the dynamic changes over time in data or task distribution (McMahan et al., 2017; Dong et al., 2022).
- There are various overlapping degrees between ordered clustering distributions $[c_1^{\mathrm{U},k}, ..., c_i^{\mathrm{U},k}, ...]$ over $\mathcal{D}^{\mathrm{U},k}$ among different participants when applying clustering algorithms locally, i.e.,

$$0 < \mathcal{P}(c_i^{\mathrm{U},k} \neq c_i^{\mathrm{U},k+1}) < 1, \quad 0 < \mathcal{P}(c_i^{\mathrm{U},k} = c_j^{\mathrm{U},k+1}) < 1, (i \neq j). \tag{6}$$

- Existing novel class number estimation methods (Han et al., 2021; Vaze et al., 2022; Chiaroni et al., 2022) require labeled data, either from known classes or some auxiliary novel classes. This

---

[1]Old participants can directly apply the data memory that is previously used to store the known class data to store the screened novel data after removing the known class data due to limited storage memory or moving them to somewhere else for privacy or intellectual property protection (Dong et al., 2022; Rebuffi et al., 2017).

requirement violates cost minimization and privacy regulations for old participants, while new participants don't have any labeled data (Dong et al., 2022). Besides, the effectiveness of these methods relies on sufficient observation of the novel data distribution, in other words, they need to access a large amount of novel class data.

**Detailed Design of `PPM`.** In order to achieve an accurate global novel class number estimation, all participants need to identify potential local prototypes within their local data first. Unlike studies (Li et al., 2023; An et al., 2022) that employ Sinkhorn-Knopp-based clustering (Genevay et al., 2019) to cluster unlabeled data into clusters with an equal size, we adopt standard Kmeans (Hartigan & Wong, 1979) to construct local prototypes. The reason is that, in FL, local data is often class-imbalanced (Wang et al., 2021). Then specifically, each participant $k$ feeds its local data $\mathcal{D}^{\mathrm{U},k}$ into the model $m^{\mathrm{L}}$ to extract representations $\boldsymbol{z}_i = f_\theta(\boldsymbol{x}_i^{\mathrm{U},k})$ and then employs Kmeans clustering to group these representations into $C^{\mathrm{L}}$ clusters[2]. Subsequently, the centroids of these clusters are utilized as local prototypes:

$$\mathcal{Z}^{\mathrm{U},k} = \left\{ \frac{\sum \mathcal{Z}_c^{\mathrm{U},k}}{|\mathcal{Z}_c^{\mathrm{U},k}|} \right\}_{c=1}^{C^{\mathrm{L}}}, \ \mathcal{Z}_c^{\mathrm{U},k} = \left\{ \boldsymbol{z} \in f_\theta(\mathcal{D}^{\mathrm{U},k}) \middle| \mathrm{Kmeans}(\boldsymbol{z}) = \boldsymbol{c}_c^{\mathrm{U},k} \right\}. \tag{7}$$

Note that these Kmeans centroids are synthetic points that achieve the minimum aggregated intra-cluster distance. In other words, they don't correspond to any real data. Besides, these local prototypes are sent to the server only once. In these ways, privacy is protected. More discussions about why our design can protect privacy are provided in the appendix.

The server shuffles all received local prototypes and constructs a prototype pool $\mathcal{Z}^{\mathrm{U}} = \cup_{k=1}^{K^{\mathrm{U}}} \mathcal{Z}^{\mathrm{U},k}$. When building `PPM`, we totally depart from conventional design principles in existing novel class merging methods Han et al. (2021); Vaze et al. (2022); Chiaroni et al. (2022). Instead, we leverage an intuition that data representations belonging to the same novel class appear in close proximity, forming high-density regions. Identifying these regions is achievable since we don't conduct any normalization when building local prototypes. Specifically, we modify DBSCAN (Schubert et al., 2017) with a rising radius $\epsilon$ and a fixed minimum cluster data size $n_{\mathrm{size}}$ to find the high-density regions. $\epsilon$ controls how large the nearby area should be considered for a certain data point, while $n_{\mathrm{size}}$ defines the requirement that if a data point is a core point, it should be surrounded by $n_{\mathrm{size}} - 1$ points. In order to cover as many high-density regions as possible, we try to use a loose setting with $n_{\mathrm{size}} = 2$. Note that setting a loose $n_{\mathrm{size}} = 2$ also considers the global imbalance case in which some data classes are minor when compared with other classes globally (Wang et al., 2021). As for $\epsilon$, we first calculate the pair-wise Euclidean distance range within the prototype pool $\mathcal{Z}^{\mathrm{U}}$ and set a rising $\epsilon$ as:

$$\{\epsilon_e = \min D + \frac{e}{E} \cdot (\max D - \min D)\}_{e=0}^E, \quad D = \{d(\boldsymbol{z}_i^{\mathrm{U}}, \boldsymbol{z}_j^{\mathrm{U}})\}_{\boldsymbol{z}_i^{\mathrm{U}}, \boldsymbol{z}_j^{\mathrm{U}} \in \mathcal{Z}^{\mathrm{U}}}, \tag{8}$$

where $d(\cdot)$ computes the Euclidean distance, and $E = 50$ is the total rising steps. For each $\epsilon_e$, we conduct DBSCAN clustering on $\mathcal{Z}^{\mathrm{U}}$ and record the unique resulting cluster number $\tilde{c}_e^{\mathrm{U}}$. After conducting the final step, we choose the maximum cluster number as the estimated global novel class number $\widetilde{C}^{\mathrm{U}} = \max \{\tilde{c}_e^{\mathrm{U}}\}_{e=0}^E$. Then we apply Kmeans again on $\mathcal{Z}^{\mathrm{U}}$ with the cluster number $\widetilde{C}^{\mathrm{U}}$ and use the cluster centroids $\hat{\boldsymbol{z}}_c^{\mathrm{U}}$ to initialize the global novel prototypes as $\mathcal{P}^{\mathrm{U}} = \{\hat{\boldsymbol{z}}_c^{\mathrm{U}}\}_{c=1}^{\widetilde{C}^{\mathrm{U}}}$. We believe that ***`PPM` can work effectively with much fewer requirements than existing methods, thus tackling the aforementioned challenges of estimating the novel class number in `FedNovel`***.

### 3.3.3 SEMANTIC-WEIGHTED LOSS

After obtaining the initial global prototypes $\mathcal{P}^{\mathrm{U}}$, instead of randomly initializing classifier $g_\omega$, we leverage $\mathcal{P}^{\mathrm{U}}$ to initialize the corresponding novel class neuron weights of $g_\omega$. Then, by viewing our problem here as weakly-supervised learning, we propose a novel *Semantic-Weighted Loss* (`SWL`) to align local data with the global prototypes $\mathcal{P}^{\mathrm{U}}$. Specifically, instead of assigning data samples with a single pseudo label, `SWL` considers the possible semantic relationships between data samples with all global prototypes, which are measured by the Softmax probability. Moreover, as $\mathcal{P}^{\mathrm{U}}$ is obtained by a Euclidean distance-based clustering algorithm (Kmeans), we directly minimize the Euclidean

---

[2]We assume that $C^{\mathrm{U}} < C^{\mathrm{L}}$ as previous studies (Zhang et al., 2022; Roy et al., 2022). Clustering local novel data into $C^{\mathrm{L}}$ clusters can cover as complete novel label space as possible.

distance between samples and prototypes. In this case, SWL is shaped as:

$$\mathcal{L}_{\mathrm{SWL}}^{\mathrm{U},k} = \frac{1}{N_B^{\mathrm{U},k}} \sum_{i=1}^{N_B^{\mathrm{U},k}} \sum_{c=1}^{\widetilde{C}^{\mathrm{U}}} \frac{\exp(f_\theta(\boldsymbol{x}_i) \cdot \hat{\boldsymbol{z}}_c^{\mathrm{U}}/\tau)}{\sum\limits_{j=1}^{\widetilde{C}^{\mathrm{U}}} \exp(f_\theta(\boldsymbol{x}_i) \cdot \hat{\boldsymbol{z}}_j^{\mathrm{U}}/\tau)} d(f_\theta(\boldsymbol{x}_i), \hat{\boldsymbol{z}}_c^{\mathrm{U}}). \tag{9}$$

By weighting the Euclidean distance with the semantic similarity, SWL can guide the model to learn more distinguishable semantic information and be more resistant to pseudo-label noise.

**Forgetting Compensation.** To preserve the knowledge learned from labeled known classes and optimize for novel classes at the same time, each client $k$ also applies a strategy of *Exponential Moving Average* (EMA) (Zhang et al., 2022) to update $f_\theta^k$ for each local training round:

$$\theta^k = \beta\theta^{\mathrm{L}} + (1-\beta)\theta^k, \tag{10}$$

where $\beta$ is a hyper-parameter to control the merging speed, which is set as $0.99$ (please see the appendix for sensitivity analysis). Note that neuron weights of classifier $g_\omega$ corresponding to known classes are frozen during GAL, while the neuron weights corresponding to novel classes are updated freely through back-propagation on the local side and FedAvg on the global side.

## 4 EXPERIMENTAL RESULTS

The major experimental setup, results, and analysis are introduced below. Please refer to the appendix and Supplementary Materials for more details and all implementation codes.

**Datasets.** *CIFAR-100* (Krizhevsky et al., 2009) consists of 100 classes, and each class contains 600 images in the size of $32 \times 32$. *Tiny-ImageNet* (Le & Yang, 2015) contains 200 classes with 500 images per class, and each image is $64 \times 64$. *ImageNet-Subset* is a subset of ImageNet (Deng et al., 2009), including 100 classes and each class has 600 images in the size of $56 \times 56$.

**Implementation details.** The number of participants is 10, and 5 participants are randomly selected as the clients to conduct 10-epoch local training in every global round. The number of global rounds for both $\mathcal{T}^{\mathrm{L}}$ and $\mathcal{T}^{\mathrm{U}}$ is 30. To simulate the non-IID setting, the data distributions of all participants follow a Dirichlet distribution (Hsu et al., 2019) with the parameter $\alpha = 0.1$ (experiments with other values can be found in the appendix). We use ResNet-18 (He et al., 2016) as the feature extractor for CIFAR-100 and Tiny-ImageNet, and ResNet-34 for ImageNet-Subset, and apply a linear layer without bias as the classifier for all datasets. The batch size is 256 and the SGD learning rate is 0.05. All experiments are conducted repeatedly with seeds 2023, 2024, and 2025.

**Evaluation metrics.** For known classes, we use ground-truth labels to calculate the classification accuracy. As for novel classes, we follow Zhang et al. (2022) to adopt Hungarian clustering accuracy (Kuhn, 1955) as the metric. We also use the overall accuracy (Roy et al., 2022) calculated on both known and novel classes to measure the model performance further. Note that the accuracy forgetting on known classes after $\mathcal{T}^{\mathrm{U}}$ of the experiments below can be found in the appendix.

**Comparison baselines.** As we are the first to address NCDL in FL, there is no baseline that is built on the same settings. For a fair comparison, we try our best to implement recent NCDL methods (AutoNovel (Han et al., 2021), GM (Zhang et al., 2022), iNCD (Roy et al., 2022), IIC (Li et al., 2023), OpenCon (Sun & Li, 2022)), novel class estimation approaches (MACC (Vaze et al., 2022), EMaCS (Chiaroni et al., 2022)), federated self-supervised learning method (Orchestra (Lubana et al., 2022)), and popular FL algorithms (FedProx (Li et al., 2020b), SCAFFOLD (Karimireddy et al., 2020), Moon (Li et al., 2021a)) in the setting of FedNovel, and compare them with our GAL.

### 4.1 EFFECTIVENESS OF GAL ON FEDNOVEL

**Comparison on one novel class learning stage.** We first assume that there is only one $\mathcal{T}^{\mathrm{U}}$ in FedNovel, which is consistent with conventional NCDL studies. For all three datasets, we set the novel class number as 20, while the rest classes are automatically regarded as known classes. All experiment results are shown in Table 1. As the novel class number 20 can be accurately estimated by PPM, we don't include it in the table to save space. The results in the table clearly demonstrate that

Table 1: Performance comparison between GAL and other baseline methods and ablation studies for FedNovel, and there is only one novel class learning stage. Left precision is the mean accuracy on certain classes, and the right is the variance of three runs. We bold and blue **the best performance**, and bold **the second-best** (not including the ablation studies).

| Method | CIFAR-100 | | | Tiny-ImageNet | | | ImageNet-Subset | | |
|---|---|---|---|---|---|---|---|---|---|
| | known | novel | all | known | novel | all | known | novel | all |
| AutoNovel | $67.3_{\pm1.0}$ | $33.4_{\pm2.4}$ | $60.6_{\pm0.5}$ | $52.3_{\pm0.4}$ | $25.0_{\pm0.7}$ | $49.6_{\pm0.4}$ | $55.1_{\pm0}$ | $21.0_{\pm0.7}$ | $48.3_{\pm0}$ |
| GM | $\mathbf{71.4_{\pm0}}$ | $29.9_{\pm0.9}$ | $\mathbf{63.1_{\pm0}}$ | $\mathbf{57.3_{\pm0}}$ | $19.4_{\pm1.2}$ | $\mathbf{53.6_{\pm0}}$ | $\mathbf{\color{blue}55.8_{\pm0}}$ | $19.9_{\pm0.9}$ | $48.6_{\pm0}$ |
| iNCD | $69.8_{\pm0.2}$ | $\mathbf{35.2_{\pm15.5}}$ | $62.9_{\pm1.2}$ | $56.4_{\pm0}$ | $\mathbf{25.3_{\pm2.1}}$ | $53.3_{\pm0}$ | $55.5_{\pm0.2}$ | $24.6_{\pm9.1}$ | $\mathbf{49.3_{\pm0.8}}$ |
| IIC | $60.8_{\pm2.3}$ | $33.1_{\pm1.8}$ | $55.3_{\pm2.4}$ | $42.8_{\pm0.7}$ | $23.3_{\pm0.6}$ | $40.9_{\pm0.5}$ | $42.3_{\pm1.5}$ | $19.1_{\pm8.2}$ | $37.7_{\pm2.4}$ |
| OpenCon | $61.5_{\pm1.7}$ | $18.4_{\pm1.4}$ | $52.9_{\pm1.5}$ | $47.6_{\pm2.0}$ | $15.6_{\pm1.7}$ | $44.4_{\pm2.0}$ | $47.4_{\pm3.9}$ | $18.0_{\pm10.1}$ | $41.6_{\pm1.0}$ |
| Orchestra | $66.6_{\pm0.5}$ | $33.1_{\pm0.8}$ | $60.0_{\pm0.2}$ | $53.4_{\pm0.3}$ | $24.7_{\pm1.1}$ | $50.1_{\pm0.1}$ | $52.8_{\pm0.3}$ | $21.5_{\pm0.3}$ | $48.4_{\pm0}$ |
| GAL-w/o $\mathcal{L}_{PCL}$ | $69.5_{\pm1.1}$ | $41.4_{\pm1.0}$ | $63.9_{\pm0.7}$ | $56.4_{\pm0}$ | $26.5_{\pm1.3}$ | $53.4_{\pm0.6}$ | $55.1_{\pm0.4}$ | $27.3_{\pm2.5}$ | $49.7_{\pm1.9}$ |
| GAL-w/o $\mathcal{L}_{SWL}$ | $72.6_{\pm0}$ | $44.7_{\pm0.1}$ | $67.0_{\pm0}$ | $57.5_{\pm0}$ | $32.1_{\pm2.3}$ | $55.0_{\pm0}$ | $55.8_{\pm0}$ | $25.0_{\pm26.8}$ | $49.6_{\pm1.1}$ |
| GAL-w/o init | $72.6_{\pm0}$ | $21.0_{\pm3.4}$ | $62.3_{\pm0.1}$ | $57.5_{\pm0}$ | $19.2_{\pm1.0}$ | $53.7_{\pm0}$ | $55.7_{\pm0}$ | $20.0_{\pm1.8}$ | $48.6_{\pm0.1}$ |
| GAL-w/o EMA | $69.6_{\pm1.1}$ | $42.4_{\pm0.5}$ | $64.2_{\pm0.8}$ | $56.0_{\pm0}$ | $29.1_{\pm1.9}$ | $53.9_{\pm1.6}$ | $55.0_{\pm1.4}$ | $27.0_{\pm0.6}$ | $49.6_{\pm0.8}$ |
| GAL | $\mathbf{\color{blue}72.6_{\pm0}}$ | $\mathbf{\color{blue}45.8_{\pm1.4}}$ | $\mathbf{\color{blue}67.3_{\pm0.1}}$ | $\mathbf{\color{blue}57.6_{\pm0}}$ | $\mathbf{\color{blue}32.7_{\pm1.7}}$ | $\mathbf{\color{blue}55.0_{\pm0}}$ | $\mathbf{\color{blue}55.8_{\pm0}}$ | $\mathbf{\color{blue}29.7_{\pm0.2}}$ | $\mathbf{\color{blue}50.5_{\pm0}}$ |

Table 2: Performance comparison between GAL and other baseline methods for FedNovel when there are two novel class learning stages with various numbers of novel classes. We only compare GAL with GM and iNCD as only they have dedicated designs for multiple NCDL stages. Novel class number estimation methods - MACC, EMaCS, and PPM, are attached to GM, iNCD, and GAL.

| Method | # Est. Method | CIFAR-100 | | | | Tiny-ImageNet | | | | ImageNet-Subset | | | |
|---|---|---|---|---|---|---|---|---|---|---|---|---|---|
| | | est.# | known | novel | all | est.# | known | novel | all | est.# | known | novel | all |
| **Novel Class Learning Stage 1: (CIFAR100: 20, Tiny-ImageNet: 30, ImageNet-Subset: 15)** | | | | | | | | | | | | | |
| GM | MACC | 6 | $\mathbf{\color{blue}72.4_{\pm0}}$ | $17.9_{\pm4.1}$ | $60.3_{\pm0.2}$ | 12 | $\mathbf{\color{blue}59.8_{\pm0}}$ | $16.9_{\pm0.7}$ | $52.6_{\pm0}$ | 6 | $54.8_{\pm0}$ | $18.2_{\pm0.2}$ | $49.1_{\pm0}$ |
| | EMaCS | 23 | $\mathbf{\color{blue}72.4_{\pm0}}$ | $21.2_{\pm0.2}$ | $61.0_{\pm0}$ | 25 | $59.7_{\pm0}$ | $19.8_{\pm4.5}$ | $53.0_{\pm0.1}$ | 34 | $54.8_{\pm0}$ | $16.6_{\pm0.5}$ | $48.8_{\pm0}$ |
| | PPM | 19 | $\mathbf{\color{blue}72.4_{\pm0}}$ | $22.0_{\pm1.9}$ | $61.2_{\pm0.1}$ | **31** | $59.7_{\pm0}$ | $19.0_{\pm2.0}$ | $52.9_{\pm0.1}$ | 12 | $54.8_{\pm0}$ | $21.0_{\pm2.9}$ | $49.5_{\pm0.1}$ |
| iNCD | MACC | 6 | $70.7_{\pm0}$ | $5.0_{\pm0}$ | $56.1_{\pm0}$ | 12 | $57.8_{\pm0}$ | $20.0_{\pm10.2}$ | $51.5_{\pm0.5}$ | 6 | $\mathbf{\color{blue}55.0_{\pm1.0}}$ | $13.5_{\pm16.8}$ | $48.4_{\pm1.7}$ |
| | EMaCS | 23 | $69.3_{\pm0.1}$ | $28.6_{\pm14.2}$ | $60.3_{\pm0.8}$ | 25 | $57.6_{\pm0}$ | $22.5_{\pm1.4}$ | $51.7_{\pm0}$ | 34 | $54.9_{\pm0.5}$ | $\mathbf{30.1_{\pm3.5}}$ | $51.0_{\pm0.7}$ |
| | PPM | 19 | $68.9_{\pm0.3}$ | $28.9_{\pm24.3}$ | $60.0_{\pm1.0}$ | **31** | $57.6_{\pm0}$ | $21.8_{\pm3.7}$ | $51.6_{\pm0.1}$ | 12 | $54.8_{\pm0.3}$ | $28.7_{\pm10.0}$ | $50.6_{\pm1.1}$ |
| GAL | MACC | 6 | $\mathbf{\color{blue}72.4_{\pm0}}$ | $25.9_{\pm0.1}$ | $62.0_{\pm0}$ | 12 | $59.6_{\pm0}$ | $24.6_{\pm0.3}$ | $53.8_{\pm0}$ | 6 | $54.8_{\pm0}$ | $23.7_{\pm5.1}$ | $49.9_{\pm0.1}$ |
| | EMaCS | 23 | $72.3_{\pm0}$ | $\mathbf{42.6_{\pm3.5}}$ | $65.7_{\pm0.2}$ | 25 | $59.6_{\pm0}$ | $\mathbf{31.0_{\pm0.5}}$ | $\mathbf{54.8_{\pm0}}$ | 34 | $54.7_{\pm0}$ | $24.1_{\pm3.3}$ | $49.9_{\pm0.1}$ |
| | PPM | 19 | $72.3_{\pm0}$ | $\mathbf{\color{blue}43.1_{\pm1.7}}$ | $\mathbf{\color{blue}65.8_{\pm0.1}}$ | **31** | $59.7_{\pm0}$ | $\mathbf{\color{blue}32.7_{\pm1.3}}$ | $\mathbf{\color{blue}55.2_{\pm0}}$ | 12 | $54.8_{\pm0}$ | $\mathbf{\color{blue}33.6_{\pm13.1}}$ | $\mathbf{\color{blue}51.4_{\pm0.3}}$ |
| **Novel Class Learning Stage 2: (CIFAR100: 10, Tiny-ImageNet: 20, ImageNet-Subset: 5)** | | | | | | | | | | | | | |
| GM | MACC | 5 | $\mathbf{\color{blue}72.4_{\pm0}}$ | $14.2_{\pm1.9}$ | $54.9_{\pm0.2}$ | 8 | $\mathbf{\color{blue}59.8_{\pm0}}$ | $9.4_{\pm20.5}$ | $47.2_{\pm1.2}$ | 4 | $\mathbf{\color{blue}54.9_{\pm0}}$ | $12.9_{\pm1.4}$ | $46.5_{\pm0}$ |
| | EMaCS | 17 | $\mathbf{\color{blue}72.4_{\pm0}}$ | $17.6_{\pm0.1}$ | $56.0_{\pm0}$ | 45 | $59.7_{\pm0}$ | $14.0_{\pm1.2}$ | $48.3_{\pm0.1}$ | 21 | $54.8_{\pm0}$ | $12.8_{\pm0.1}$ | $46.4_{\pm0}$ |
| | PPM | **9** | $72.3_{\pm0}$ | $17.8_{\pm0.4}$ | $56.0_{\pm0}$ | **19** | $\mathbf{\color{blue}59.8_{\pm0}}$ | $13.4_{\pm0.4}$ | $48.2_{\pm0}$ | **5** | $\mathbf{\color{blue}54.9_{\pm0}}$ | $15.6_{\pm4.6}$ | $47.0_{\pm0.2}$ |
| iNCD | MACC | 5 | $70.0_{\pm0}$ | $3.3_{\pm0}$ | $50.0_{\pm0}$ | 8 | $56.9_{\pm0.1}$ | $7.3_{\pm21.4}$ | $44.5_{\pm1.3}$ | 4 | $53.4_{\pm3.4}$ | $12.0_{\pm3.3}$ | $45.1_{\pm2.4}$ |
| | EMaCS | 17 | $68.0_{\pm0.2}$ | $19.5_{\pm1.9}$ | $53.5_{\pm0.1}$ | 45 | $55.9_{\pm1.2}$ | $9.1_{\pm6.3}$ | $44.5_{\pm0.4}$ | 21 | $54.2_{\pm0.4}$ | $17.3_{\pm1.0}$ | $46.8_{\pm0.2}$ |
| | PPM | **9** | $67.8_{\pm0.3}$ | $17.9_{\pm1.3}$ | $52.8_{\pm0.5}$ | **19** | $56.9_{\pm0.2}$ | $9.5_{\pm3.7}$ | $45.0_{\pm0.1}$ | **5** | $52.6_{\pm0.9}$ | $8.8_{\pm1.8}$ | $43.8_{\pm0.3}$ |
| GAL | MACC | 5 | $\mathbf{\color{blue}72.4_{\pm0}}$ | $24.8_{\pm0}$ | $58.1_{\pm0}$ | 8 | $59.6_{\pm0}$ | $\mathbf{19.0_{\pm0.5}}$ | $\mathbf{49.5_{\pm0}}$ | 4 | $54.7_{\pm0}$ | $14.8_{\pm1.3}$ | $46.8_{\pm0.1}$ |
| | EMaCS | 17 | $\mathbf{\color{blue}72.4_{\pm0}}$ | $\mathbf{35.0_{\pm6.1}}$ | $\mathbf{61.2_{\pm0.6}}$ | 45 | $59.7_{\pm0}$ | $18.5_{\pm0}$ | $49.4_{\pm0}$ | 21 | $54.7_{\pm0}$ | $\mathbf{21.2_{\pm0.4}}$ | $\mathbf{48.0_{\pm0}}$ |
| | PPM | **9** | $72.4_{\pm0}$ | $\mathbf{\color{blue}37.4_{\pm4.2}}$ | $\mathbf{\color{blue}61.9_{\pm0.4}}$ | **19** | $59.6_{\pm0}$ | $\mathbf{\color{blue}25.5_{\pm0.3}}$ | $\mathbf{\color{blue}51.1_{\pm0}}$ | **5** | $54.8_{\pm0}$ | $\mathbf{\color{blue}25.1_{\pm9.0}}$ | $\mathbf{\color{blue}48.8_{\pm0.8}}$ |

*our GAL framework always achieves the best performance on all metrics, including accuracy on known, novel, and all classes.* The performance improvement on novel class accuracy is particularly promising, ranging from 5.1% on ImageNet-Subset to 10.6% on CIFAR-100 over the best baseline method, which demonstrates the effectiveness of GAL in solving FedNovel.

**Experimental results on two novel class learning stages.** As mentioned in related works (Dong et al., 2022), FL participants often collect unseen novel classes continually. For such cases, we split each dataset into one labeled subset and two unlabeled subsets with disjoint classes, and conduct two $\mathcal{T}^U$ in sequence. The detailed class number settings for each novel stage are shown in Table 2. For a fair comparison, we only compare GAL with GM and iNCD, as only they have dedicated designs for multiple novel class learning stages. Furthermore, to validate the effectiveness of PPM, the novel class number of each stage is estimated by state-of-the-art approaches (MACC and EMaCS). All experiments are conducted using the estimated class number and the detailed results are shown in Table 2. We can observe that *our PPM provides the most accurate class number estimation at each stage for all datasets,* with only errors of $\pm1$ in most cases and $\pm3$ in the worst case. Moreover, *GAL with PPM achieves the best performance on all datasets*, showing 3.5%-14.2% improvement on novel classes over GM and iNCD in stage 1 and 7.8%-17.9% in stage 2.

Table 3: Performance evaluation of combining `GAL` with other FL algorithms. We assume that Kmeans can equip FedProx, SCAFFOLD, and Moon with a certain level of NCDL capability, while `GAL` can substantially exceed it. The `FedNovel` setting is the same as Table 1.

| Method | CIFAR-100 | | | Tiny-ImageNet | | | ImageNet-Subset | | |
|---|---|---|---|---|---|---|---|---|---|
| | known | novel | all | known | novel | all | known | novel | all |
| FedProx+Kmeans | $69.8_{\pm 0}$ | $28.0_{\pm 0.4}$ | $61.5_{\pm 0.2}$ | $56.3_{\pm 0.2}$ | $13.2_{\pm 0.5}$ | $52.1_{\pm 0.3}$ | $66.1_{\pm 0}$ | $11.5_{\pm 1.1}$ | $55.2_{\pm 0.8}$ |
| FedProx+GAL | $69.8_{\pm 0}$ | $\mathbf{40.1}_{\pm 2.0}$ | $63.9_{\pm 1.4}$ | $56.4_{\pm 0}$ | $\mathbf{28.9}_{\pm 0.6}$ | $53.7_{\pm 0.5}$ | $66.2_{\pm 0.4}$ | $\mathbf{28.0}_{\pm 1.4}$ | $58.5_{\pm 1.3}$ |
| SCAFFOLD+Kmeans | $71.0_{\pm 0}$ | $25.5_{\pm 1.1}$ | $61.6_{\pm 0.8}$ | $53.0_{\pm 0.5}$ | $10.9_{\pm 0.9}$ | $48.7_{\pm 0.5}$ | $58.7_{\pm 0.2}$ | $11.8_{\pm 2.0}$ | $52.2_{\pm 1.4}$ |
| SCAFFOLD+GAL | $71.2_{\pm 0}$ | $\mathbf{45.2}_{\pm 1.7}$ | $65.8_{\pm 1.3}$ | $54.5_{\pm 0}$ | $\mathbf{31.1}_{\pm 1.0}$ | $53.2_{\pm 0.9}$ | $59.0_{\pm 1.0}$ | $\mathbf{30.4}_{\pm 0.7}$ | $54.9_{\pm 0.2}$ |
| Moon+Kmeans | $71.9_{\pm 0}$ | $31.7_{\pm 0.7}$ | $63.7_{\pm 0.5}$ | $57.5_{\pm 0}$ | $22.0_{\pm 1.5}$ | $54.0_{\pm 1.0}$ | $55.8_{\pm 0.2}$ | $10.5_{\pm 1.4}$ | $46.7_{\pm 0.9}$ |
| Moon+GAL | $72.0_{\pm 0}$ | $\mathbf{46.1}_{\pm 0.7}$ | $66.9_{\pm 0.2}$ | $57.5_{\pm 0}$ | $\mathbf{33.5}_{\pm 1.0}$ | $55.1_{\pm 0.4}$ | $56.2_{\pm 0}$ | $\mathbf{30.8}_{\pm 0.1}$ | $50.8_{\pm 0}$ |

Figure 2: Effectiveness of using global prototypes to initialize the classifier neuron weights for baseline methods. The experiments are conducted in `FedNovel` with one novel class learning stage, which is exactly the same as Table 1. Only novel class accuracy is reported.

**Experiments with more FL algorithms.** To further evaluate `GAL`, we also carry out experiments combining `GAL` with FedProx, SCAFFOLD, and Moon. Specifically, we assume that regular FL can possess a certain level of NCDL capability through unsupervised clustering (e.g., Kmeans) on novel class data representations. The experiment results are shown in Table 3. We can see that ***using `GAL` with these FL algorithms provides much better performance on novel classes.***

**More experiments in the appendix.** In addition to the above experiments, please refer to the appendix for: 1) experiment results in the centralized setting, 2) experiment results with various degrees of heterogeneity in `FedNovel`, 3) experiment results in settings with different data partitions, 4) experiment results on fine-grained datasets (CUB200, StanfordCars, and Herbarium 19), 5) experiment results of `PPM` in settings of various novel class numbers, 6) sensitivity analysis of hyperparameters, 7) experiment results with a large number of participants, and 8) experiment results of launching data reconstruction attacks for verifying `GAL`'s characteristic of privacy-preserving.

## 4.2 Ablation Study

We investigate the importance of each component in `GAL` via an ablation study. As shown in Table 1, '`GAL`-w/o $\mathcal{L}_{\mathrm{PCL}}$', '`GAL`-w/o $\mathcal{L}_{\mathrm{SWL}}$', '`GAL`-w/o init', '`GAL`-w/o EMA' denote the performance of `GAL` without using PCL in $\mathcal{T}^{\mathrm{L}}$, without using `SWL` (replacing it with a Binary CrossEntropy Loss (Han et al., 2021)), without initializing the classifier with global prototypes, and without applying EMA (replacing it with distillation (Roy et al., 2022)), respectively. We can observe that these versions of `GAL` degrade evidently, which shows the importance of all these modules in our approach. We also equip other baseline methods with the classifier initialization from global prototypes, which seems to be the most critical module in `GAL`. From the results shown in Figure 2, global prototype-induced initialization also brings evident performance improvement for those baselines.

## 5 Conclusion

In this paper, we study an important and practical problem called *Federated Novel Class Learning* (`FedNovel`), and propose *Global Alignment Learning* (`GAL`) to help the FL model discover and learn novel classes. `GAL` can not only accurately estimate the novel class number and construct corresponding global prototypes, but also guide the local training of different participants to tackle data heterogeneity. Extensive experiments demonstrate that `GAL` significantly outperforms state-of-the-art novel class learning methods in various cases for the `FedNovel` problem.

ETHICS STATEMENT

In this paper, our work is not related to human subjects, practices to data set releases, discrimination/bias/fairness concerns, and also do not have legal compliance or research integrity issues. Our work is proposed to enable federated learning (FL) systems to discover and learn unseen novel data classes when deploying in real-world applications. In this case, if the FL systems are used responsibly for good purposes, we believe that our proposed methods will not cause ethical issues or pose negative societal impacts.

REPRODUCIBILITY STATEMENT

The source code is provided in the supplementary materials. All datasets we use are public. In addition, we also provide detailed experiment parameters and random seeds in the appendix.

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

# SUMMARY OF THE APPENDIX

This appendix contains additional details for the ICLR 2024 submission "*FedNovel: Federated Novel Class Learning*", including implementation details, additional experiment results, discussion of limitations, and broader impact. The implementation code can be found in the supplementary materials. The appendix is organized as follows:

- Section A introduces the data splitting strategy (Section A.1), details of the baseline implementation (Section A.2), and detailed settings of the ablation study and experiment results when there is a mixture between known and novel classes during novel class learning (Section A.3).

- Section B provides additional results for 1) experiments in the centralized setting (Section B.1); 2) experiments with various degrees of heterogeneity in `FedNovel` (Section B.2); 3) experiments in settings with different data partitions (Section B.3); 4) experiments on fine-grained datasets (CUB200, StanfordCars, and Herbarium 19) (Section B.4); 5) experiments of `PPM` in settings of various novel class numbers (Section B.5); 6) sensitivity analysis of hyper-parameters (Section B.6); 7) experiments with a large number of participants (Section B.7); 8) experiments of launching data reconstruction attacks for verifying `GAL`'s characteristic of privacy-preserving (Section B.8); and 9) results of performance forgetting on known classes (Section B.9).

- Section C presents the detailed optimization pipeline of `GAL`.

- Section D discusses the limitations of the proposed methods and the possible future directions.

- Section E illustrates the potential broader impact of this work in the real world.

## A  IMPLEMENTATION DETAILS

### A.1  DATA SPLITTING SETTINGS

CIFAR-100 contains two separate sets – one is a training set consisting of 500 samples for each class, and the other is used for testing with 100 samples for each class. For Tiny-ImageNet and ImageNet-Subset, we follow FCIL (Dong et al., 2022) to select the ending 50 and 100 samples per class for testing, respectively, while the rest 500 samples are used for training. To simulate the practical non-IID setting, the training samples of each participant are drawn independently with class labels following a categorical distribution over $C = C^{\mathrm{L}} + C^{\mathrm{U}}$ classes, which can be parameterized by a vector $\mathbf{q}$ ($q_i \geq 0, i \in [1, C]$ and $\|\mathbf{q}\|_1 = 1$). And we draw $\mathbf{q} \sim \mathrm{Dir}(\alpha, \mathbf{p})$ from a Dirichlet Distribution (Hsu et al., 2019), where $\mathbf{p}$ is a prior class distribution over $C$ classes and $\alpha > 0$ controls the data heterogeneity among FL participants. A smaller $\alpha$ leads to more heterogeneous data distributions among participants, and we set $\alpha = 0.1$ for all `FedNovel` experiments in the main paper, while the centralized training setting can be regarded as the case when $\alpha \to +\infty$.

### A.2  BASELINE METHODS

To our best knowledge, this paper is the first to tackle the problem of `FedNovel`, and thus there is no direct baseline method that can be used to compare with our proposed methods. In this case, for a fair comparison, we try our best to integrate certain state-of-the-art baseline methods from related areas, like standard novel class learning and federated self-supervised learning, into FL to solve `FedNovel`. In addition to standard FedAvg (McMahan et al., 2017), we also implement other mainstream FL algorithms including FedProx (Li et al., 2020b), SCAFFOLD (Karimireddy et al., 2020), and Moon (Li et al., 2021a), to see whether our `GAL` can empower them the capability of NCDL. Next, we will provide the details of implementing these state-of-the-art baselines.

First of all, we assume that all baseline methods rely on the same $m^{\mathrm{L}}$ trained by `GAL` at the beginning of novel class learning. AutoNovel (Han et al., 2021) assumes that both the known and the novel class data are accessible during novel class learning, and it has separate design and training loss for known and novel data. Considering the unavailability of the known class data in `FedNovel`, we

only apply its pair-wise Binary CrossEntropy (BCE) loss without using the labeled known data:

$$\mathcal{L}_{\text{BCE}}^{\text{U},k} = -\frac{1}{N_B^{\text{U}^2}} \sum_{i=1}^{N_B^{\text{U}}} \sum_{j=1}^{N_B^{\text{U}}} \left( s_{ij} \log g_\omega^{\text{U}}(\boldsymbol{z}_i^{\text{U},k})^\top g_\omega^{\text{U}}(\boldsymbol{z}_j^{\text{U},k}) + (1 - s_{ij}) \log(1 - g_\omega^{\text{U}}(\boldsymbol{z}_i^{\text{U},k})^\top g_\omega^{\text{U}}(\boldsymbol{z}_j^{\text{U},k})) \right),$$

(11)

where $g_\omega^{\text{U}}$ is the novel head of classifier. $s_{ij}$ is obtained by using feature rank statistics, and $s_{ij} = 1$ when the top-k ranked dimensions of two samples in a data pair are identical, otherwise $s_{ij} = 0$. Besides, when we apply AutoNovel in the centralized training setting, we observe that the model performance drops significantly due to the forgetting of learned knowledge in known classes. To compensate for such forgetting, we also equip AutoNovel with EMA with $\beta = 0.99$.

GM (Zhang et al., 2022) is proposed to incrementally discover and learn novel classes, and thus its problem settings are similar to `FedNovel` except for whether the model is trained via FL or not. In the design of GM, the model is alternatively updated by a growing phase and a merging phase using the novel class data only. In the setting of `FedNovel`, each selected client conducts the growing phase in the first 20 global epochs and then conducts the merging phase in the last 10 global epochs. Like `GAL`, the EMA is applied after each global epoch, and $\beta$ is set as 0.99. iNCD (Roy et al., 2022) is also designed for discovering and learning novel classes continually like GM. There is a feature replay loss in iNCD that relies on the statistical information of known class data, and we provide such information during conducting iNCD in FL, though we believe that it is unreasonable for both the server and clients to be aware of statistics of known classes. IIC (Li et al., 2023) heavily relies on known data to construct the loss function, and thus we have to randomly pick half of the known data and provide them to IIC in the novel class learning stage. OpenCon (Sun & Li, 2022), as another label-available approach, is provided with the full set of known class data during the stage of novel class learning in `FedNovel`. As IIC and OpenCon have been provided with known class data, we don't incorporate forgetting compensation techniques into them. As for Orchestra (Lubana et al., 2022), although it is a federated self-supervised learning method, only a few modifications are needed to adapt it for solving `FedNovel`. Specifically, we first apply Orchestra to tune the feature extractor on unlabeled novel class data. During the tuning, we equip Orchestra with EMA to preserve the known class performance. After sufficient tuning (30 global rounds for each novel class learning stage), the feature extractor is expected to produce more distinguishable representations for novel class data. At this moment, we freeze the feature extractor followed by a new classifier head, and train this head with pair-wise BCE loss of AutoNovel.

In addition to these NCDL baseline methods, we also provide the implementation details of two state-of-the-art novel class number estimation approaches (MACC (Vaze et al., 2022) and EMaCS (Chiaroni et al., 2022)) that have been compared with our proposed `PPM` (Potential Prototype Merge) in experiments of two novel class learning stages. As mentioned in the main paper, existing novel class number estimation methods usually require a certain number of labeled data – MACC and EMaCS are no exceptions. Fortunately, MACC and EMaCS operate in a way that doesn't need labeled data from novel classes. Instead, they only require labeled data from known classes. And this requirement can be met without the need to directly provide real known class data. Specifically, we apply prototype augmentations on known class prototypes $\mathcal{P}^{\text{L}}$ constructed as neuron weights of classifier $g_\omega$ to get labeled known class representations as follows:

$$\{\boldsymbol{z}_{c,i} = \boldsymbol{p}_c^{\text{L}} + n * \min d(\mathcal{Z}^{\text{U}})\}_{i=1,c=1}^{K^{\text{U}},C^{\text{U}}},$$

(12)

where $n \sim \mathcal{N}(0, 0.01)$ is a random vector with the same dimension as prototypes (we have tried our best to find a suitable Gaussian distribution and found the one with a variance of $0.01$ is the best), and $d(\cdot)$ computes all sample-pair Euclidean distances. The reason why we use the minimum sample-pair distance in the local prototype pool $\mathcal{Z}^{\text{U}}$ is to ensure that the augmented representations are located in clusters with high density.

Then, we also provide the details of implementing the combination methods between other FL algorithms and Kmeans or our `GAL`. FedProx introduces a regularization term for balancing the difference between local models and the global model at each round. Thus we follow FedProx to add such regularization to both known class learning and novel class learning stages. As for SCAFFOLD, it incorporates a control variate that indicates the stage of each FL participant, and with this control variate, the drift caused by non-IID can be mitigated. In the problem of our interest, we maintain the updating policy of this control variate in SCAFFOLD for both $\mathcal{T}^{\text{L}}$ and $\mathcal{T}^{\text{U}}$. Moon leverages the

principle of contrastive learning and proposes to conduct model parameter contrastive comparison against non-IID. We also include such parameter contrastive loss in the training of both $\mathcal{T}^L$ and $\mathcal{T}^U$. As mentioned in the main paper, unsupervised clustering, such as Kmeans, can be applied to these FL algorithms to discover and learn novel classes. But note that dedicated modifications are still needed as unsupervised clustering is impacted by non-IID, e.g., Kmeans is non-parametric thus it is hard to develop a global Kmeans mechanism that is effective to all participants without cluster merging. To apply Kmeans, we leverage `PPM` to merge clusters of participants and build global prototypes. Then we share these global prototypes with all participants, and they can calculate the distance between their local samples and these prototypes to allocate the cluster labels. On the other hand, the combination between `GAL` and these FL algorithms is quite simple and straightforward. For example, the modification in $\mathcal{T}^L$ is adding PCL $\mathcal{L}_{PCL}$ to the training loss. As for $\mathcal{T}^U$, we only need to apply `PPM` to initialize the classifier parameters for novel classes before the training and then use `SWL` to train the FL model.

### A.3 ABLATION STUDY SETTINGS

According to the detachability and fungibility of different modules in `GAL`, we conduct a thorough ablation study that has been mentioned in the main paper. Here we provide detailed experiment settings and more results. To validate the effectiveness of `SWL`, instead of arbitrarily replacing `SWL` with some loss functions that are expected to perform poorly, e.g., pseudo-labeling-based CrossEntropy loss or prototype learning loss, we choose the most effective self-training loss in label-unavailable NCDL – pair-wise BCE loss (Han et al., 2021). EMA is used in `GAL` to compensate for the forgetting of known classes during the process of novel class learning. We replace EMA with the old model logits distillation and known prototype augmentation-based feature replay, which are used in iNCD (Roy et al., 2022) to alleviate forgetting on known classes, during the ablation study. As for verifying the effectiveness of global prototype-induced classifier initialization, we randomly initialize the neuron weights of the novel classifier head with a normal distribution instead. Moreover, as we believe that our global prototypes can bring general benefits when dealing with data heterogeneity in FL, we also equip other baseline methods with our global prototype-induced classifier initialization, which has been reported in the main paper. In addition to Figure 2 in the main paper, we provide the detailed results in Table 4 including the accuracies on known and all classes. We also validate the representation enhancement of our modified prototype contrastive loss by detaching it from the training of known class learning stage in the ablation study.

**Experiment when there is a mixture of both novel and known classes in the unlabeled data.** In the current design of `GAL`, we apply a data filtering mechanism to screen out novel class data and don't use unlabeled known class data during novel class learning. However, it only requires a minor modification if we want to handle the scenario where there is a mixture of both unlabeled known and novel class data. That is, we can leverage the data filtering mechanism to separate known and novel class data, and after that, apply softmax to conduct pseudo label learning on known class data while conducting `GAL` on novel class data. We carry out experiments for such case, and the results are shown in the row 'GAL-mixture' of Table 4. According to these results, there is nearly no influence when incorporating known class data in the training. Certainly, we will explore better utilization of unlabeled known class data in the future.

## B MORE EXPERIMENTS

### B.1 EXPERIMENT RESULTS IN THE CENTRALIZED SETTING

To further evaluate the general application potential of `GAL`, we also carry out experiments in a centralized training setting. Specifically, we assume that there is only one participant that owns the entire dataset conducting one novel class learning stage with 20 novel classes, and the results are shown in Table 5. It is clear that ***GAL achieves the best performance on nearly all metrics even in the centralized learning setting.*** In addition, we also conduct an ablation study in the centralized setting. The experiment results in Table 5 show that the full `GAL` performs the best, indicating that every module of our approach plays an important role in facilitating better novel class learning, even for non-FL scenarios.

Table 4: Performance comparison between GAL and other baselines equipped with initialization from global prototypes for FedNovel. The ending 20 classes are the novel classes and there is only one novel class learning stage. Such experiment results validate the importance of global prototypes and the effectiveness of initializing the classifier with these prototypes.

| Method | CIFAR-100 | | | Tiny-ImageNet | | | ImageNet-Subset | | |
|---|---|---|---|---|---|---|---|---|---|
| | known | novel | all | known | novel | all | known | novel | all |
| AutoNovel+init | $71.4_{\pm0}$ | $43.7_{\pm0.9}$ | $65.9_{\pm0}$ | $57.3_{\pm0}$ | $32.1_{\pm5.1}$ | $54.8_{\pm0.1}$ | $\mathbf{55.8}_{\pm0}$ | $15.5_{\pm11.7}$ | $47.7_{\pm0.5}$ |
| GM+init | $71.4_{\pm0}$ | $38.1_{\pm15.3}$ | $64.7_{\pm0.7}$ | $57.2_{\pm0}$ | $29.2_{\pm14.3}$ | $54.4_{\pm0.2}$ | $56.0_{\pm0}$ | $14.2_{\pm2.6}$ | $47.7_{\pm0.1}$ |
| iNCD+init | $71.5_{\pm0.1}$ | $36.9_{\pm13.3}$ | $64.6_{\pm0.3}$ | $57.1_{\pm0}$ | $28.5_{\pm4.2}$ | $54.2_{\pm0.1}$ | $55.7_{\pm0.3}$ | $26.3_{\pm0}$ | $49.8_{\pm0.2}$ |
| IIC+init | $62.8_{\pm4.9}$ | $37.0_{\pm8.0}$ | $57.7_{\pm4.9}$ | $44.1_{\pm0.3}$ | $27.5_{\pm0.5}$ | $42.4_{\pm0.2}$ | $48.9_{\pm0.1}$ | $29.4_{\pm1.6}$ | $45.0_{\pm0}$ |
| Orchestra+init | $66.7_{\pm0.4}$ | $34.8_{\pm2.3}$ | $60.2_{\pm0.2}$ | $53.0_{\pm0.7}$ | $26.3_{\pm0.4}$ | $50.1_{\pm0}$ | $52.7_{\pm0.2}$ | $23.5_{\pm0.3}$ | $48.6_{\pm0}$ |
| OpenCon+init | $61.7_{\pm2.4}$ | $42.0_{\pm3.1}$ | $56.9_{\pm0.1}$ | $47.6_{\pm2.0}$ | $28.7_{\pm6.0}$ | $45.7_{\pm1.9}$ | $47.4_{\pm3.9}$ | $22.9_{\pm3.1}$ | $42.5_{\pm2.3}$ |
| GAL-mixture | $\mathbf{72.6}_{\pm0}$ | $45.4_{\pm1.0}$ | $67.2_{\pm0.5}$ | $57.5_{\pm0}$ | $32.2_{\pm0.4}$ | $54.9_{\pm0}$ | $55.5_{\pm0}$ | $\mathbf{30.0}_{\pm1.1}$ | $\mathbf{50.5}_{\pm0.3}$ |
| GAL | $\mathbf{72.6}_{\pm0}$ | $\mathbf{45.8}_{\pm1.4}$ | $\mathbf{67.3}_{\pm0.1}$ | $\mathbf{57.6}_{\pm0}$ | $\mathbf{32.7}_{\pm1.7}$ | $\mathbf{55.0}_{\pm0}$ | $\mathbf{55.8}_{\pm0}$ | $29.7_{\pm0.2}$ | $\mathbf{50.5}_{\pm0}$ |

Table 5: Performance comparison between GAL and other NCDL methods in centralized training setting. The ending 20 classes are the novel classes and there is only one novel class learning stage.

| Method | CIFAR-100 | | | Tiny-ImageNet | | | ImageNet-Subset | | |
|---|---|---|---|---|---|---|---|---|---|
| | known | novel | all | known | novel | all | known | novel | all |
| AutoNovel | $34.6_{\pm1.4}$ | $42.0_{\pm8.4}$ | $36.1_{\pm2.0}$ | $27.2_{\pm1.5}$ | $30.7_{\pm0.9}$ | $27.5_{\pm1.5}$ | $41.5_{\pm0.3}$ | $31.9_{\pm7.3}$ | $39.5_{\pm0.2}$ |
| GM | $71.5_{\pm0}$ | $32.0_{\pm3.0}$ | $63.6_{\pm0.1}$ | $57.3_{\pm0}$ | $25.7_{\pm1.6}$ | $54.2_{\pm0}$ | $\mathbf{55.8}_{\pm0}$ | $26.9_{\pm8.4}$ | $50.0_{\pm0.3}$ |
| iNCD | $66.3_{\pm1.6}$ | $40.9_{\pm7.7}$ | $61.2_{\pm2.5}$ | $53.0_{\pm1.4}$ | $29.0_{\pm0.5}$ | $50.6_{\pm1.2}$ | $51.7_{\pm0.2}$ | $31.8_{\pm2.1}$ | $47.8_{\pm0.2}$ |
| IIC | $68.2_{\pm1.0}$ | $45.4_{\pm3.9}$ | $63.6_{\pm1.1}$ | $55.5_{\pm0.3}$ | $27.0_{\pm1.3}$ | $52.6_{\pm0.3}$ | $52.1_{\pm1.1}$ | $30.4_{\pm3.4}$ | $47.7_{\pm0.2}$ |
| OpenCon | $63.3_{\pm1.0}$ | $30.7_{\pm9.7}$ | $56.8_{\pm0.9}$ | $48.4_{\pm0.3}$ | $29.9_{\pm0}$ | $46.2_{\pm0}$ | $47.2_{\pm4.2}$ | $22.6_{\pm2.4}$ | $42.3_{\pm3.6}$ |
| GAL-w/o $\mathcal{L}_{PCL}$ | $71.5_{\pm0}$ | $46.5_{\pm0.4}$ | $67.3_{\pm0}$ | $56.5_{\pm0}$ | $36.2_{\pm3.0}$ | $\mathbf{55.4}_{\pm0.7}$ | $55.0_{\pm0}$ | $30.4_{\pm2.3}$ | $50.9_{\pm0.9}$ |
| GAL-w/o $\mathcal{L}_{SWL}$ | $\mathbf{72.6}_{\pm0}$ | $46.7_{\pm4.4}$ | $67.4_{\pm0.2}$ | $\mathbf{57.5}_{\pm0}$ | $36.0_{\pm13.0}$ | $\mathbf{55.4}_{\pm0.1}$ | $55.7_{\pm0}$ | $29.4_{\pm7.5}$ | $50.5_{\pm0.3}$ |
| GAL-w/o init | $\mathbf{72.6}_{\pm0}$ | $39.6_{\pm4.2}$ | $66.0_{\pm0.2}$ | $\mathbf{57.5}_{\pm0}$ | $27.2_{\pm0.3}$ | $54.5_{\pm0}$ | $55.7_{\pm0}$ | $22.1_{\pm1.9}$ | $49.0_{\pm0.1}$ |
| GAL-w/o EMA | $71.5_{\pm0}$ | $46.6_{\pm3.8}$ | $66.6_{\pm0.1}$ | $55.5_{\pm0}$ | $36.5_{\pm5.9}$ | $53.6_{\pm0.1}$ | $55.0_{\pm0}$ | $30.0_{\pm3.9}$ | $50.0_{\pm0.3}$ |
| GAL | $\mathbf{72.6}_{\pm0}$ | $\mathbf{47.7}_{\pm2.9}$ | $\mathbf{67.6}_{\pm0.1}$ | $\mathbf{57.5}_{\pm0}$ | $\mathbf{36.7}_{\pm6.5}$ | $\mathbf{55.4}_{\pm0.1}$ | $55.7_{\pm0}$ | $\mathbf{32.2}_{\pm0.5}$ | $\mathbf{51.0}_{\pm0}$ |

## B.2 EXPERIMENT RESULTS WITH VARIOUS DEGREES OF HETEROGENEITY IN FEDNOVEL

It is worth exploring how robust GAL is when faced with varying levels of data heterogeneity. Therefore, we test GAL and other baseline methods in a more challenging FL scenario, in which the data distributions among participants are more heterogeneous ($\alpha$ of Dirichlet Distribution is set as $0.001$). From the experiment results in Table 6, we can observe that GAL still performs the best in all cases on all metrics. Combined with the experiment results shown in Tables 1 and 5, which exactly correspond to the settings of $\alpha = 0.1$ and $\alpha \to +\infty$, we can conclude that our approach GAL is able to consistently perform well and achieve effective novel class learning in FedNovel under various levels of data heterogeneity.

Table 6: Performance comparison between GAL and other baselines for FedNovel with more heterogeneous data distributions ($\alpha$ of Dirichlet Distribution is set as $0.001$). The ending 20 classes are the novel classes with only one novel class learning stage. The experiment results present that GAL can retain the performance superiority when the data heterogeneity becomes much heavier.

| Method | CIFAR-100 | | | Tiny-ImageNet | | | ImageNet-Subset | | |
|---|---|---|---|---|---|---|---|---|---|
| | known | novel | all | known | novel | all | known | novel | all |
| AutoNovel | $65.8_{\pm4.6}$ | $28.8_{\pm7.0}$ | $58.4_{\pm4.5}$ | $34.7_{\pm14.0}$ | $23.0_{\pm8.1}$ | $33.3_{\pm11.8}$ | $54.6_{\pm0.1}$ | $20.4_{\pm3.0}$ | $47.8_{\pm0.1}$ |
| GM | $71.5_{\pm0}$ | $27.3_{\pm0.9}$ | $62.7_{\pm0}$ | $57.3_{\pm0}$ | $19.6_{\pm3.7}$ | $53.6_{\pm0}$ | $55.8_{\pm0}$ | $19.4_{\pm0.7}$ | $47.5_{\pm2.7}$ |
| iNCD | $69.9_{\pm0.1}$ | $31.5_{\pm3.7}$ | $62.2_{\pm0}$ | $56.2_{\pm0.4}$ | $6.5_{\pm7.1}$ | $51.2_{\pm0.7}$ | $56.0_{\pm0.3}$ | $21.3_{\pm1.1}$ | $48.9_{\pm0}$ |
| IIC | $66.8_{\pm0.5}$ | $20.8_{\pm6.6}$ | $57.6_{\pm1.2}$ | $47.7_{\pm0.9}$ | $15.6_{\pm2.1}$ | $44.5_{\pm1.1}$ | $53.0_{\pm0}$ | $15.3_{\pm0}$ | $45.4_{\pm0}$ |
| Orchestra | $66.9_{\pm0.1}$ | $30.2_{\pm0.8}$ | $58.2_{\pm0}$ | $53.0_{\pm0.1}$ | $22.4_{\pm0.5}$ | $48.7_{\pm0.2}$ | $52.2_{\pm0.4}$ | $19.7_{\pm0.1}$ | $46.6_{\pm0}$ |
| OpenCon | $20.3_{\pm13.6}$ | $22.1_{\pm12.0}$ | $20.6_{\pm12.9}$ | $11.9_{\pm2.7}$ | $16.5_{\pm10.7}$ | $12.6_{\pm4.3}$ | $20.9_{\pm7.1}$ | $17.6_{\pm0.3}$ | $20.2_{\pm5.0}$ |
| GAL | $\mathbf{72.6}_{\pm0}$ | $\mathbf{36.2}_{\pm5.0}$ | $\mathbf{65.3}_{\pm0.2}$ | $\mathbf{57.5}_{\pm0}$ | $\mathbf{29.7}_{\pm0.5}$ | $\mathbf{54.7}_{\pm0}$ | $\mathbf{55.7}_{\pm0}$ | $\mathbf{22.9}_{\pm3.3}$ | $\mathbf{49.2}_{\pm0.1}$ |

## B.3 Experiment results in settings with different data partitions

In experiments of the main paper, following regular NCDL studies (Roy et al., 2022; Zhang et al., 2022), we choose the same widely-used data partition settings based on ordered label sequence, but this does not mean that the effectiveness of PPM relies on the data partition. We conduct additional experiments of various data partitioning settings by randomly selecting 20 classes as the novel ones while the rest are the known classes with three seeds, 2023, 2024, and 2025, respectively. The detailed results are shown in Tables 7, 8 and 9. We can observe that regardless of data partitions, PPM always provides accurate novel class number estimation, and GAL performs the best all the time.

Table 7: Performance comparison between GAL and other baselines for FedNovel with different data partitions. 20 classes of CIFAR-100 are randomly selected with different seeds as the novel classes with one novel class learning stage. The experiment results present that both PPM and GAL are robust to different data partitions.

| Seed | 2023 | | | 2024 | | | 2025 | | |
|---|---|---|---|---|---|---|---|---|---|
| PPM est.# | 19 | | | 20 | | | 20 | | |
| Method | known | novel | all | known | novel | all | known | novel | all |
| AutoNovel | 73.1 | 21.3 | 62.7 | 73.1 | 22.1 | 63.0 | 71.5 | 21.6 | 61.8 |
| GM | 71.1 | 38.3 | 64.5 | 70.4 | 21.6 | 60.6 | 69.7 | 41.5 | 64.1 |
| iNCD | 73.0 | 24.7 | 63.4 | 72.5 | 25.5 | 63.3 | 71.0 | 24.4 | 62.6 |
| IIC | 55.1 | 36.2 | 51.3 | 49.8 | 33.1 | 46.4 | 51.6 | 36.4 | 48.6 |
| OpenCon | 60.7 | 20.9 | 52.8 | 61.5 | 18.7 | 52.9 | 60.0 | 19.5 | 52.2 |
| Orchestra | 71.1 | 22.8 | 61.5 | 72.0 | 25.5 | 62.8 | 70.0 | 20.8 | 60.8 |
| GAL | **73.1** | **48.5** | **68.2** | **73.3** | **46.3** | **67.9** | **71.9** | **43.0** | **66.1** |

Table 8: Performance comparison between GAL and other baselines for FedNovel with different data partitions. 20 classes of Tiny-ImageNet are randomly selected with different seeds as the novel classes with one novel class learning stage. The experiment results present that both PPM and GAL are robust to different data partitions.

| Seed | 2023 | | | 2024 | | | 2025 | | |
|---|---|---|---|---|---|---|---|---|---|
| PPM est.# | 21 | | | 19 | | | 20 | | |
| Method | known | novel | all | known | novel | all | known | novel | all |
| AutoNovel | 56.5 | 24.9 | 53.5 | 56.2 | 20.6 | 53.1 | 57.1 | 19.7 | 53.4 |
| GM | 56.6 | 22.0 | 53.1 | 56.7 | 24.8 | 53.5 | 56.2 | 26.2 | 53.2 |
| iNCD | 56.5 | 27.8 | 53.8 | 56.0 | 22.7 | 53.3 | 56.5 | 22.4 | 53.3 |
| IIC | 46.5 | 23.1 | 44.2 | 48.2 | 21.7 | 45.6 | 47.3 | 18.6 | 44.4 |
| OpenCon | 45.5 | 16.0 | 43.2 | 42.7 | 18.2 | 40.1 | 47.0 | 15.5 | 43.8 |
| Orchestra | 56.5 | 25.0 | 53.5 | 56.5 | 22.4 | 53.4 | 56.7 | 22.0 | 53.3 |
| GAL | **56.7** | **46.8** | **55.7** | **56.8** | **34.9** | **54.7** | **57.3** | **35.8** | **55.2** |

## B.4 Experiment results on fine-grained datasets

To further evaluate the application potential in a wider range, we also conduct experiments on three fine-grained datasets: CUB200 (Wah et al., 2011), StanfordCars (Krause et al., 2013), and Herbarium 19 (Tan et al., 2019), which are often used in regular NCDL studies. For these three datasets, we downscale the image size to 64×64 and apply ResNet-34 as the feature extractor. We follow regular NCDL works to set their known class numbers as 160, 130, and 600 respectively (accordingly, with novel class numbers 20, 20, and 83 respectively). We conduct the experiments with only one novel class learning stage, and the results are shown in Table 10. We can observe that GAL still performs the best on all three fine-grained datasets in all cases.

Table 9: Performance comparison between `GAL` and other baselines for `FedNovel` with different data partitions. 20 classes of ImageNet-Subset are randomly selected with different seeds as the novel classes with one novel class learning stage. The experiment results present that both `PPM` and `GAL` are robust to different data partitions.

| Seed | 2023 | | | 2024 | | | 2025 | | |
|------|------|------|-----|------|------|-----|------|------|-----|
| `PPM` est.# | 19 | | | 19 | | | 20 | | |
| Method | known | novel | all | known | novel | all | known | novel | all |
| AutoNovel | 54.0 | 19.5 | 47.1 | 57.2 | 22.9 | 50.4 | 55.5 | 17.0 | 47.8 |
| GM | 54.0 | 29.7 | 49.3 | 55.3 | 25.2 | 49.3 | 54.7 | 24.1 | 48.6 |
| iNCD | 54.0 | 22.7 | 47.6 | 56.7 | 23.0 | 50.1 | 55.0 | 20.7 | 48.1 |
| IIC | 54.0 | 27.7 | 48.8 | 54.2 | 30.6 | 49.5 | 54.3 | 26.6 | 48.7 |
| OpenCon | 47.0 | 16.6 | 41.2 | 46.7 | 18.3 | 41.4 | 47.2 | 18.0 | 41.5 |
| Orchestra | 53.0 | 20.7 | 47.0 | 56.2 | 24.0 | 50.1 | 55.0 | 20.5 | 48.1 |
| GAL | **54.1** | **35.4** | **50.4** | **57.2** | **33.4** | **52.5** | **55.6** | **30.7** | **50.6** |

Table 10: Performance comparison between `GAL` and other baselines for `FedNovel` with three fine-grained dataset. Only one novel class learning stage is used and the novel class number is 20, 20, and 83 for CUB200, StanfordCars, and Herbarium 19 respectively. The experiment results present that `GAL` can retain the performance superiority on these three fine-grained datasets.

| Method | CUB200 | | | StanfordCars | | | Herbarium 19 | | |
|--------|--------|------|-----|--------------|------|-----|--------------|------|-----|
| | known | novel | all | known | novel | all | known | novel | all |
| AutoNovel | $40.0_{\pm0}$ | $18.1_{\pm2.3}$ | $37.5_{\pm0.5}$ | $45.0_{\pm0}$ | $18.2_{\pm4.2}$ | $42.2_{\pm1.1}$ | $49.2_{\pm0}$ | $21.7_{\pm1.1}$ | $45.4_{\pm0.2}$ |
| GM | $38.5_{\pm2.2}$ | $19.7_{\pm5.5}$ | $37.3_{\pm2.0}$ | $45.0_{\pm0}$ | $12.5_{\pm3.1}$ | $41.6_{\pm1.7}$ | $49.2_{\pm0}$ | $22.7_{\pm1.5}$ | $45.5_{\pm0.9}$ |
| iNCD | $39.2_{\pm1.0}$ | $20.0_{\pm2.0}$ | $37.2_{\pm1.2}$ | $43.1_{\pm1.7}$ | $19.0_{\pm1.5}$ | $42.0_{\pm1.0}$ | $48.5_{\pm0.7}$ | $22.4_{\pm0.9}$ | $45.2_{\pm0}$ |
| IIC | $38.7_{\pm1.1}$ | $18.5_{\pm0.9}$ | $36.6_{\pm0.4}$ | $33.3_{\pm4.7}$ | $22.6_{\pm2.2}$ | $32.2_{\pm2.0}$ | $49.2_{\pm0}$ | $17.8_{\pm2.5}$ | $44.9_{\pm1.6}$ |
| OpenCon | $29.9_{\pm3.4}$ | $11.2_{\pm6.7}$ | $28.3_{\pm2.5}$ | $34.7_{\pm2.9}$ | $14.3_{\pm7.0}$ | $31.5_{\pm3.7}$ | $35.5_{\pm4.4}$ | $13.4_{\pm8.9}$ | $33.2_{\pm3.9}$ |
| Orchestra | $40.0_{\pm0}$ | $18.5_{\pm1.0}$ | $37.5_{\pm0.1}$ | $44.5_{\pm0.5}$ | $18.3_{\pm2.0}$ | $42.3_{\pm0.4}$ | $48.0_{\pm1.3}$ | $22.0_{\pm0.9}$ | $45.2_{\pm0.5}$ |
| GAL | $\mathbf{40.5_{\pm0}}$ | $\mathbf{25.3_{\pm2.4}}$ | $\mathbf{39.5_{\pm1.0}}$ | $\mathbf{45.0_{\pm0}}$ | $\mathbf{25.0_{\pm1.3}}$ | $\mathbf{43.6_{\pm0.5}}$ | $\mathbf{49.3_{\pm0.2}}$ | $\mathbf{30.4_{\pm2.8}}$ | $\mathbf{46.8_{\pm1.1}}$ |

## B.5 EXPERIMENT RESULTS OF `PPM` IN SETTINGS OF VARIOUS NOVEL CLASS NUMBERS

As discussed in the main paper, the class number of novel data is critical for effective novel class learning, and it is challenging to acquire and estimate in `FedNovel`. To address this issue, we propose the `PPM` to search for high-density regions in aggregated local prototypes. Different from existing class number estimation approaches (AutoNovel (Han et al., 2021), MACC (Vaze et al., 2022), EMaCS (Chiaroni et al., 2022)), `PPM` does not require any labeled data, from either known classes or novel classes. For a fair comparison, we apply representation augmentation on known class prototypes to generate labeled representations for MACC and EMaCS instead of providing real data. We don't compare AutoNovel (Han et al., 2021) with `PPM` since AutoNovel requires additional labeled novel data, which is too strong to be assumed in the setting of `FedNovel`. To comprehensively assess the effectiveness of these methods for estimating class numbers, we conduct additional experiments by varying the number of novel classes to be estimated. Specifically, we set different novel class numbers starting from 5 with an increased stride of 5 for CIFAR-100 and ImageNet-Subset and set novel class numbers starting from 10 with a stride of 10 for Tiny-ImageNet, CUB200, StanfordCars, and Herbarium 19. Based on the experiment results in Tables 11 and 12, `PPM` always provides the most accurate estimation of the novel class number, showing its consistent effectiveness.

## B.6 SENSITIVITY ANALYSIS OF HYPER-PARAMETERS

There are several hyper-parameters in our algorithm, some of which are directly adopted from common values. For instance, the $\tau$ in Eqs. 4 and 9 are set to 0.07 as the standard contrastive loss for fair comparison. As for others, we conduct thorough sensitivity analysis, including $\eta$ in the known class learning stage and $\beta$ in EMA on CIFAR-100 with one novel class learning stage. The experi-

Table 11: Performance comparison between `PPM` and other novel class number estimation approaches in settings of various novel class numbers. The experiments here show that `PPM` can consistently provide the most accurate novel class number estimation for cases of different class numbers.

| Dataset | CIFAR-100 | | | | | | Tiny-ImageNet | | | | | ImageNet-Subset | | | | Average |
|---|---|---|---|---|---|---|---|---|---|---|---|---|---|---|---|---|
| True Class # | 5 | 10 | 15 | 20 | 25 | 30 | 10 | 20 | 30 | 40 | 50 | 5 | 10 | 15 | 20 | Error (%) |
| MACC | 4 | 5 | 6 | 6 | 14 | 15 | 6 | 8 | 12 | 22 | 28 | 4 | 6 | 6 | 14 | 46.2 |
| EMaCS | 18 | 17 | 30 | 23 | 12 | 11 | 27 | 45 | 25 | 26 | 58 | 21 | 20 | 34 | 15 | 99.6 |
| PPM | **4** | **9** | **13** | **19** | **25** | **29** | **8** | **19** | **31** | **40** | **54** | **5** | **8** | **12** | **17** | **9.5** |

Table 12: Performance comparison between `PPM` and other novel class number estimation approaches on three fine-grained datasets in settings of various novel class numbers. The experiments here show that `PPM` can consistently provide the most accurate novel class number estimation for cases of different class numbers.

| Dataset | CUB200 | | | | StanfordCars | | | | Herbarium 19 | | | | Average |
|---|---|---|---|---|---|---|---|---|---|---|---|---|---|
| True Class # | 10 | 20 | 30 | 40 | 10 | 20 | 30 | 40 | 20 | 40 | 60 | 80 | Error (%) |
| MACC | 4 | 6 | 13 | 18 | 6 | 12 | 20 | 24 | 9 | 18 | 24 | 30 | 52.3 |
| EMaCS | 21 | 35 | 34 | 29 | 19 | 45 | 42 | 36 | 25 | 17 | 38 | 46 | 54.4 |
| PPM | **9** | **21** | **32** | **44** | **11** | **20** | **28** | **39** | **19** | **42** | **65** | **86** | **6.4** |

ment results are shown in Table 13. We can observe that `GAL` is robust to different values of $\eta$, and performs the best when $\eta = 0.10$. Different values of $\beta$ directly associate with the preservation of feature extraction ability learned in the known class learning stage, thus it impacts the performance of `GAL`.

Table 13: Sensitivity analysis of $\eta$ in known class training stage and $\beta$ in EMA. Experiments are conducted using CIFAR-100 with one novel class learning stage (20 novel classes).

| $\eta$ | known | novel | all | $\beta$ | known | novel | all |
|---|---|---|---|---|---|---|---|
| 0.02 | 71.5 | 44.4 | 66.1 | 0.10 | 40.7 | 24.0 | 34.3 |
| 0.05 | 71.8 | 44.9 | 66.4 | 0.50 | 59.6 | 31.4 | 54.0 |
| 0.10 | **72.6** | **45.8** | **67.3** | 0.80 | 67.9 | 37.9 | 62.1 |
| 0.50 | 70.9 | 44.5 | 65.9 | 0.95 | 71.9 | 42.5 | 66.7 |
| 1.00 | 71.0 | 44.7 | 66.0 | 0.99 | **72.6** | **45.8** | **67.3** |

### B.7 EXPERIMENT RESULTS WITH A LARGE NUMBER OF PARTICIPANTS

In real-world application scenarios, there can be a large amount of participants in the FL system. Thus, we evaluate `FedNovel` under a situation where there are a large number of participants for CIFAR-100 with one novel class learning stage. Specifically, we randomly select 20 clients from all participants every global round to conduct local training. The results are shown in Table 14, which shows that `GAL` can still perform the best when there are many participants.

### B.8 EXPERIMENT RESULTS OF LAUNCHING DATA RECONSTRUCTION ATTACKS

In `GAL`, the sharing of local prototypes only occurs once before novel class learning, which is more secure than many FL works (Tan et al., 2022; Huang et al., 2023) of other fields where prototypes are shared every round. The local prototypes are constructed by conducting unsupervised clustering on unlabeled novel class data. Each participant maintains an identical count of clusters, and the semantic affiliations of distinct clusters across different clients diverge, as illustrated in Eq. 6 of the main text. Therefore, a direct comparison of cluster labels of local prototypes does not divulge label information. Label information leakage could arise from comparing the similarities between different participants' local prototypes. However, when local prototypes are shared, the model's capability to extract meaningful features from unlabeled novel class data is relatively weak. Consequently, the similarity between prototypes is unreliable, implying that similar prototypes might correspond to different classes, and distinct prototypes could potentially correspond to the same class. This also prevents privacy leakage caused by data reconstruction attacks. The reason is that when constructing prototypes, each cluster contains a mixture of multiple classes due to the weak feature extraction

Table 14: Performance comparison between `GAL` and other baselines for `FedNovel` with more participants. The ending 20 classes of CIFAR-100 are used as the novel classes and there is only one novel class learning stage in `FedNovel`. The experiment results present that `GAL` can retain the performance superiority where there are a large number of participants in `FedNovel`.

| Participant # | 20 | | | 50 | | | 100 | | |
|---|---|---|---|---|---|---|---|---|---|
| Method | known | novel | all | known | novel | all | known | novel | all |
| AutoNovel | 68.2 | 30.1 | 60.5 | 68.0 | 27.9 | 59.6 | 68.3 | 27.0 | 59.5 |
| GM | 71.1 | 36.1 | 64.1 | 71.1 | 31.3 | 63.2 | 71.1 | 30.8 | 63.1 |
| iNCD | 70.5 | 31.7 | 62.7 | 70.5 | 29.6 | 62.3 | 71.1 | 25.2 | 62.0 |
| IIC | 61.8 | 21.7 | 53.8 | 62.3 | 22.4 | 54.6 | 61.1 | 16.2 | 52.1 |
| Orchestra | 67.7 | 31.1 | 60.6 | 68.5 | 25.6 | 59.3 | 67.8 | 26.0 | 59.2 |
| OpenCon | 61.7 | 21.0 | 53.6 | 71.9 | 17.8 | 61.1 | 71.7 | 19.3 | 61.2 |
| GAL | **72.6** | **45.4** | **67.3** | **72.6** | **44.9** | **67.1** | **72.6** | **42.5** | **66.6** |

ability, thereby causing the cluster centers to contain information from multiple classes. Moreover, for those clusters only with small data volumes, the high similarity in the representation space may not reliably correspond to a similarly high similarity in the input space. Moreover, works (Tan et al., 2022; Huang et al., 2023) highlight that prototypes are formed in a low-dimensional space by averaging data representations, and mainstream model structures include numerous operations of dropout, pooling, and ReLU activations. These two factors are irreversible, which further strengthens the privacy preservation of our work.

Given the absence of data reconstruction attacks against representations in the literature, we speculate that attacks are most likely to follow Deepleakage (Zhu et al., 2019) – optimizing dummy data to align the representation as closely as possible with the target. The results of this attack are shown in Figure 3. It is nearly impossible to capture interpretable semantics from the recovered images, showing that the privacy of prototypes is preserved.

## B.9 EXPERIMENT RESULTS OF THE FORGETTING OF KNOWN CLASSES

To measure the forgetting of known classes after novel class learning, we only need to know the performance of $m^L$ right after known class learning. Therefore, here we provide the detailed results before $\mathcal{T}^U$ for experiments of Table 1 in the main paper, which is shown as Table 15.

Table 15: Model performance before novel class learning. All baseline methods rely on the same model $m^L$. Performance forgetting of known classes can be calculated by subtracting the known accuracy after novel class learning.

| Metric | CIFAR-100 | | | Tiny-ImageNet | | | ImageNet-Subset | | |
|---|---|---|---|---|---|---|---|---|---|
| | known | novel | all | known | novel | all | known | novel | all |
| $m^L$ | $72.7_{\pm 0.1}$ | $13.4_{\pm 4.2}$ | $59.9_{\pm 1.5}$ | $57.6_{\pm 0}$ | $15.7_{\pm 3.1}$ | $53.4_{\pm 1.2}$ | $55.8_{\pm 0.1}$ | $12.8_{\pm 2.0}$ | $47.2_{\pm 1.1}$ |

## C OVERALL OPTIMIZATION PIPELINE

`GAL` is proposed to enable FL systems to discover and learn unseen novel classes. We assume any FL system can periodically leverage `GAL` to incorporate novel classes or functionalities after the training on labeled known class data. Specifically, when the novel class learning stage starts, all available FL participants at that moment first need to apply unsupervised clustering on their local data to find local potential prototypes. These local prototypes will be sent to the server only once and then `GAL` will apply `PPM` to merge these prototypes into global novel prototypes. After initializing the novel classifier head with the global prototypes, at each round, each selected client can leverage `SWL` to conduct local training. This pipeline is shown in Algorithm 1, and the overall `GAL` workflow is Algorithm 2.



CIFAR-100

Tiny-ImageNet

ImageNet-Subset

Figure 3: Recovered images of conducting data reconstruction attack on three local prototypes of three datasets. The data reconstruction attack optimizes the recovered images in the input space and tries to make their representations as close to local prototypes as possible. We apply Adam to minimize the mean square errors until the error cannot decrease further.

---

**Algorithm 1** Novel Class Learning.

**Given:** After training on known class data, a model $m = f_\theta \circ g_\omega$ is given as the basis of novel class learning. At this moment, there are $K^{\mathrm{U}}$ active participants $\{\mathcal{S}^1, \mathcal{S}^2, ..., \mathcal{S}^{K^{\mathrm{U}}}\}$, and each holds its unlabeled novel class dataset $\{\mathcal{D}^{\mathrm{U},1}, \mathcal{D}^{\mathrm{U},2}, ..., \mathcal{D}^{\mathrm{U},K^{\mathrm{U}}}\}$. PPM ascending total iteration is $E$. FL total training round is $E_g$.

**All Active Participants:**

**for** $\mathcal{S}^k$ *in* $\{\mathcal{S}^1, \mathcal{S}^2, \cdots, \mathcal{S}^{K^{\mathrm{U}}}\}$ **do**
    Apply Kmeans on $f_\theta(\mathcal{D}^{\mathrm{U},k})$;
    Return Kmeans cluster centers $\mathcal{Z}^{\mathrm{U},k}$;

**Central Server:**

// Conduct PPM
Receive and shuffle $\mathcal{Z}^{\mathrm{U}} = \cup_{k=1}^{K^{\mathrm{U}}} \mathcal{Z}^{\mathrm{U},k}$;
**for** $e = 0, \cdots, E$ **do**
    $D = \{d(\boldsymbol{z}_i^{\mathrm{U}}, \boldsymbol{z}_j^{\mathrm{U}})\}_{\boldsymbol{z}_i^{\mathrm{U}}, \boldsymbol{z}_j^{\mathrm{U}} \in \mathcal{Z}^{\mathrm{U}}}$;
    $\epsilon_e = \min D + \frac{e}{E} \cdot (\max D - \min D)$;
    Conduct DBSCAN with $n_{\mathrm{size}} = 2$ and $\epsilon_e$;
    Record unique cluster number $\tilde{c}_e^{\mathrm{U}}$;

Obtain # of novel class as $\widetilde{C}^{\mathrm{U}} = \max\{\tilde{c}_e^{\mathrm{U}}\}_{e=0}^{E}$;
Apply Kmeans with $\widetilde{C}^{\mathrm{U}}$ to construct global prototypes $\mathcal{P}^{\mathrm{U}} = \{\hat{\boldsymbol{z}}_c^{\mathrm{U}}\}_{c=1}^{\widetilde{C}^{\mathrm{U}}}$ and initialize $g_\omega$;

**Selected Clients:**

**for** $t = 1, \cdots, E_g$ **do**
    Central Server randomly selects $\mathcal{K}^{\mathrm{U}}$ clients;
    **for** $\mathcal{S}^{s,k}$ *in* $\{\mathcal{S}^{s,1}, \mathcal{S}^{s,2}, \cdots, \mathcal{S}^{s,\mathcal{K}^{\mathrm{U}}}\}$ **do**
        Apply SWL to train $m$;
        Update $f_\theta$ in EMA and $g_\omega$ in backpropagation;
        Upload model gradients to the Central Server.
    Central Server applies FedAvg;
    Central Server distributes aggregated gradients;

---

**Algorithm 2** Overall GAL Workflow.

**Given:** A global model $m = f_\theta \circ g_\omega$ consists of a feature extractor $f_\theta$ and a classifier $g_\omega$. At the beginning, there are $K^{\mathrm{L}}$ participants $\{\mathcal{S}^1, \mathcal{S}^2, ..., \mathcal{S}^{K^{\mathrm{L}}}\}$, and each holds its labeled known class dataset $\{\mathcal{D}^{\mathrm{L},1}, \mathcal{D}^{\mathrm{L},2}, ..., \mathcal{D}^{\mathrm{L},K^{\mathrm{L}}}\}$. FL total training round is $E_g$.

**Known Class Learning:**

**for** $t = 1, \cdots, E_g$ **do**
    Central Server randomly selects $\mathcal{K}^{\mathrm{L}}$ clients;
    **for** $\mathcal{S}^{s,k}$ *in* $\{\mathcal{S}^{s,1}, \mathcal{S}^{s,2}, \cdots, \mathcal{S}^{s,\mathcal{K}^{\mathrm{L}}}\}$ **do**
        Apply $\mathcal{L}_{\mathcal{T}^{\mathrm{L}}} = \mathcal{L}_{\mathrm{CE}} + \eta \mathcal{L}_{\mathrm{PCL}}$ to train $m$;
        Upload $\nabla_{\theta,\omega} \mathcal{L}_{\mathcal{T}^{\mathrm{L}}}^k$ to the Central Server;
    Central Server applies FedAvg to calculate the aggregated model gradients $\nabla_{\theta,\omega} \mathcal{L}_{\mathcal{T}^{\mathrm{L}}}^{\mathrm{Avg}}$;
    Central Server distributes $\nabla_{\theta,\omega} \mathcal{L}_{\mathcal{T}^{\mathrm{L}}}^{\mathrm{Avg}}$ to all participants.

**Novel Class Data Filtering**:

**for** $\mathcal{S}^k$ *in* $\{\mathcal{S}^1, ..., \mathcal{S}^{\mathrm{U}}\}$ **do**
    **while** $\mathcal{D}^{\mathrm{U},k}$ *is not full* **do**
        Apply Eq. (5) to filter out known class data;
        Store the remaining data to the data memory and form the novel class dataset $\mathcal{D}^{\mathrm{U},k}$;

**Novel Class Learning**:

Conduct Algorithm 1;

## D    LIMITATIONS AND FUTURE WORK

This paper mainly focuses on achieving novel class learning for FL from the algorithmic perspective. Although the proposed GAL can be empirically demonstrated effective, there is no rigorous theoretical proof that this will always be the case. Therefore, in future work, establishing a theoretical framework for FedNovel and providing rigorous analysis of GAL could be the first step. Moreover, considering FedNovel in more advanced scenarios and domains, e.g., medical semantic segmentation and some natural language processing cases, is also worthy of exploring in the future.

## E    BROADER IMPACT

The FedNovel study outlined in this paper presents substantial societal implications and potential benefits without an apparent negative impact. As privacy concerns become increasingly important, the need for efficient methods to handle dynamic data distributions without compromising privacy is critical. Our GAL framework is designed to address the challenges of FedNovel, specifically in merging and aligning novel classes identified and learned by different clients while preserving privacy. This, in essence, supports a more sustainable and adaptable machine learning system that can evolve with changing data scenarios. GAL's impressive results, even in non-FL or centralized training scenarios, indicate its potential for wide-reaching application in various real-world scenarios. These scenarios include but are not limited to, healthcare, financial services, telecommunications, and social networking platforms, where preserving user privacy while continually adapting to new information is paramount. Moreover, the improved model performance achieved with GAL can contribute to more reliable and efficient systems, enhancing user experiences and outcomes. We believe that our research in FedNovel and the development of GAL pave the way for advancements in privacy-conscious, dynamic learning systems, fostering a more secure and adaptable digital landscape.

