# OpenReview forum: "FedNovel: Federated Novel Class Learning"
_ICLR.cc/2024/Conference — Submitted to ICLR 2024_

### Official Review · Reviewer_hsUT · 2023-10-30

**Soundness:** 3 good
**Presentation:** 2 fair
**Contribution:** 3 good
**Rating:** 5
**Confidence:** 3

**Summary:**

This paper explores novel class discovery and learning within the framework of federated learning, addressing the challenge of evolving data on local client devices. The study introduces a Global Alignment learning framework aimed at estimating the number of global novel classes and providing guidance for local training. Specifically, this framework initially estimates the number of novel classes by identifying high-density regions within the representation space. Then, it captures all potential correlations between prototypes and training data to alleviate issues related to data heterogeneity. The proposed approach demonstrates advanced performance across a range of benchmark datasets

**Strengths:**

1. The paper is well-written and easy to read.

2. It introduces an effective approach for discovering novel classes with Global Alignment learning, specifically targeting the federated learning setting with dynamic change in data distribution.

3. The authors conducted thorough experiments to validate the effectiveness of their proposed approach. Through empirical evidence, they support their claims and offer insights into the performance and advantages of novel class learning under the constraint of non-iid data, and privacy protection.

**Weaknesses:**

1. The method lacks sufficient elaboration and requires additional effort to comprehend the underlying technique. For instance, it encompasses multiple design elements, yet Figure 1 fails to offer adequate details to facilitate a clear understanding of the approach. And it is difficult to visualize the algorithm. It would be better if the paper provided an algorithm block. (Q1, Q2, Q4)

2. Some of the assertions made in the paper need additional justifications. (Q3)

**Questions:**

1. What is the training process?

(a) In section 3.2, the known classes depend on Equation (2) to converge to model $m^L$. Then Section 3.3.1 includes a modified PCL to enhance training. Is the modified PCL included during the training for $m^L$ or after the global server achieved $m^L$?

(b) From Figure 1, the local prototypes are only uploaded once to the server, which contradicts my understanding of FL: the local models should communicate with the central server for several rounds until converge. And as the local model keeps updating, the local prototypes should also evolve, how does one-time uploading transmit every local information?

2. Section 3.2 mentions the unlabeled testing datasets "belongs to a unified novel label space", does it assume the number of classes of novel samples is fixed? If this number is not fixed, then how to decide the number of clusters? If this number is fixed, how does it apply to the setting where the novel data are continually emerging?

3. Section 3.3.2 chooses neuron weights as prototypes since the data from other clients is unavailable in the FL setting. However, there is still a lot of difference between weights and features. Any justification to support that using weights as prototypes is as effective as data/features?

4. The "anchor sample" first appears in Section 3.3.1. How are they selected, or what is their definition?

5. In Section 3.3.2, there is a data memory storing filtered-out data. Are these data filtered out because they are known class data?

---

> ### Author Response · Authors · 2023-11-20
> **Response to Reviewer hsUT Part [1]**
>
> We really appreciate your constructive comments and suggestions. We provide the following response to answer your questions and address your concerns.
>
>
> Weaknesses:
>
> >1. The method lacks sufficient elaboration and requires additional effort to comprehend the underlying technique.
>
> Thanks for your suggestion. We have revised Figure 1 in our manuscript to include more information and details, enabling it to more clearly express the method we propose. As for the algorithm flowchart, we have provided the optimization pipeline for the novel class learning stage in the appendix (Section C). In our revision, we present the complete GAL algorithm process, encompassing both the known class learning and novel class learning stages.
>
> >2. Some of the assertions made in the paper need additional justifications.
>
> Thank you for your comment. We have answered your questions in detail in the subsequent Question Section. Please refer to the responses provided below.
>
> Questions:
>
> >1.(a) The known class learning training loss.
>
> We apologize for the confusion caused. To develop model $m^L$, during the known class learning, we train the model using both the modified Prototype Contrastive Learning (PCL) and cross-entropy loss simultaneously.
>
> >1.(b) The question about the prototype communication.
>
> As illustrated in Figure 1, **local prototypes are uploaded to the server only once** at the beginning of novel class learning. Upon receiving these local prototypes, the server uses Potential Prototype Merging (PPM) to estimate the number of novel classes and to construct global prototypes. These global prototypes are then distributed to all clients, who use them to initialize the novel class classifier. Then, the novel class classifier is updated through local training and central aggregation. Thus, **the update of the novel class classifier essentially equals the evolution of the global prototypes.** In this case, the entire novel class learning process requires only a single communication of local prototypes at the beginning. The reduction in the number of prototype communications further enhances privacy protection.
>
> >2. Whether the number of novel classes is fixed or not during novel class learning.
>
> In FedNovel, we assume that **novel class learning can be conducted periodically**. During each novel class learning stage, we only invite clients who are active and available at the time to participate, and we pause the update of their data memories. Therefore, in this case, the number of novel classes in each novel class learning stage remains constant. For the continuously incoming novel data, we can initiate another stage of novel class learning when the novel data memory of the majority of clients is nearly full.
>
> >3. Any justification to support that using weights as prototypes is as effective as data/features?
>
> Your understanding is correct. The weights of a standard linear fully connected layer indeed differ significantly from features. However, under specific conditions, if we ensure that **the bias of the linear fully connected layer are zero**, and train the neural network using the cross-entropy loss, the weights of the linear fully connected layer can be viewed as prototypes [1]. Moreover, the classifier consists solely of a single unbiased linear fully connected layer, which is put right after the feature extractor. Consequently, **the dimensionality of this layer's weights is the same as that of the extracted representations**. With the bias consistently set to zero, the computation of cross-entropy loss is based on the softmax probabilities derived from the inner products between the representations and the weights of that linear layer. Minimizing cross-entropy loss increases the inner product between the sample representation and the corresponding neuron weight (associated with the sample's ground truth label). The inner product between vectors is proportionally related to the cosine similarity between them. Thus, **an increase in the inner product equals an increase in cosine similarity**. Therefore, we can consider the neuron weights of the linear fully connected layer as prototypes based on cosine similarity.
>
> >4. The typo of "anchor sample" that appears in Section 3.3.1.
>
> Thanks for your question and careful reading. The term 'anchor sample' was a typo. Our original intention was to refer to each regular training sample. We have removed the term 'anchor' in our revision.

---

> > ### Author Response · Authors · 2023-11-20
> > **Response to Reviewer hsUT Part [2]**
> >
> > >5. Question about the filtering strategy of novel data memory.
> >
> > Yes, your understanding is correct. Before the formal start of novel class learning, clients' novel data memory will collect continuously incoming streaming data. The streaming data includes both known class data and potentially data from unseen novel classes. **Our current design does not utilize known class data in novel class learning**, thus we need to filter it out. Then, clients' novel data memory will store only novel class data. We have also conducted **experiments with the presence of known class data**; please refer to Section A.3 in the appendix. According to the experimental results, using known class data in the novel class learning stage does not lead to performance improvement. Therefore, to further reduce the cost of storage and computational resources, we choose to filter out known class data.
> >
> > [1] Semi-supervised domain adaptation via minimax entropy. ICCV, 2019

---

> ### Author Response · Authors · 2023-11-21
> **A Gentle Reminder of Further Feedback**
>
> Dear Reviewer hsUT,
>
> The conclusion of the discussion period is closing, and we eagerly await your response. We greatly appreciate your time and effort in reviewing this paper and helping us improve it.
>
> Thank you again for the detailed and constructive reviews. We hope our response is able to address your comments related to the training pipeline clarification, questions about the novel class number, justification about neural parameter-based prototypes, and clarification of novel data memory. We take this as a great opportunity to improve our work and shall be grateful for any additional feedback you could give us.
>
> Best Regards,
>
> Authors of Paper 5479

---

> ### Author Response · Authors · 2023-11-22
> **Discussion Phase Ends in Less Than 24 Hours. We Eagerly Await Reviewer hsUT's Feedback.**
>
> Dear Reviewer hsUT,
>
> As the rebuttal discussion phase ends in less than 24 hours, we want to express our gratitude for your engagement thus far. We shall kindly remind you that after the 22nd, we are not allowed to respond to your further questions you may have. We really want to check with you whether our response addresses your concerns during the author-reviewer discussion phase. We have diligently addressed every concern and question you raised during the initial review, and our extensive efforts are aimed at enhancing the clarity and quality of our work.
>
> Your feedback is really important to us. We eagerly await any potential updates to your ratings, as they play a critical role in the assessment of our paper. We genuinely hope our responses have resolved your concerns and provided satisfactory explanations. Your thoughtful evaluation greatly aids in our paper's refinement and strength. We sincerely appreciate your dedication and time again.
>
> Best regards,
>
> Authors of Paper 5479

---

> ### Author Response · Authors · 2023-11-23
> **Final Reminder before Reviewer-Author Discussion Phase Closure for Reviewer hsUT**
>
> Dear Reviewer hsUT,
>
> Thank you again for the initial comments. As the Reviewer-Author Discussion phase is closing in less than 12 hours (Nov. 22nd AoE), we would greatly appreciate any feedback on our rebuttal. We fully understand that you may be busy at this time, but hope that you could kindly have a quick look at our responses and assess whether they have addressed your concerns and warrant an update to the rating. We would also welcome any additional feedback and questions.
>
> Best Regards,
>
> Authors of Paper 5479

---

### Official Review · Reviewer_RSBY · 2023-10-31

**Soundness:** 2 fair
**Presentation:** 3 good
**Contribution:** 1 poor
**Rating:** 3
**Confidence:** 5

**Summary:**

This paper proposes a prototype-based class number estimation method for Federated Novel Class Discovery, where the model is required to merge and align novel classes that are discovered and learned by different clients under privacy constraint. Extensive experimental results demonstrate the effectiveness of proposed method.

**Strengths:**

1.	This paper is well-written, well-organized and easy to follow.
2.	The performance of the proposed method is impressive.
3.	Ablation studies are comprehensive and demonstrate the effectiveness of proposed method.

**Weaknesses:**

1.	Lacks of crucial reference literatures [A][B]. Thus, Federated New Class Discovery/Learning is not a new research problem. In contribution, authors say,” we are the first to focus on this problem and propose an effective solution”. It might be somewhat overclaiming.
2.	Insufficient comparison and discussion. From my understanding, the proposed method is similar to commonly-used semi-supervised learning methods.
3.	Limited novelty. From the perspective of NCD methods, prototype-based contrastive learning and low confidence sample rejection have explored in [C]. Why is the proposed method superior to [C]? From the perspective of semi-supervised federated learning methods, the federated prototype learning has been studied in [D][E]. What’s the novelty of the proposed methods compared with [D][E]?
4.	From my understanding, [A][B] can also be used in federated new class learning. It is better to discuss and compare the proposed method with [A][B].
5.	Lack of theoretical proof why the estimation method is better than other competitors.
6.	The reasonableness of the experiment setting still needs to be considered. As authors claimed in the paper, they use Dirichlet Distribution to control data heterogeneity. (1) It makes client labeled and unlabeled data highly-unbalanced. However, in [F], the labeled and unlabeled data are kept fixed partitioning. It is better to discuss the relationship between proposed method and data distribution. (2) It might lead clients to have some sharing categories or have non-overlapping categories. How does the proposed method solve both situations?

[A] Towards Unbiased Training in Federated Open-world Semi-supervised Learning. ICML, 2023.
[B] Federated Generalized Category Discovery, Arxiv, 2023.
[C] Opencon: Open-world contrastive learning. TMLR, 2022.
[D] Fedproto: Federated prototype learning across heterogeneous clients, AAAI,2022.
[E]Federated Semi-Supervised Learning with Prototypical Networks, Arxiv,2022.
[F] Generalized Category Discovery, CVPR, 2022.

**Questions:**

Please see Weaknesses.

---

> ### Author Response · Authors · 2023-11-20
> **Response to Reviewer RSBY Part [1]**
>
> We would like to thank you for your constructive comments. We will address your concerns, answer your questions, and clarify your misunderstandings as follows.
>
> >1. Lacks of crucial reference literature.
>
> To prevent misunderstandings, we need to clarify that Federated OpenSet/OpenWorld Learning (FOL) is related to but distinctly different from, the FedNovel problem we are focusing on. Firstly, **their application scenarios differ**. FOL is typically applied in static federated training, where the simultaneous learning of known and novel classes is achieved within a single training phase. In contrast, FedNovel focuses on dynamic federated training, where the federated model, after having been employed for a period following the training of known classes, is then found to require the learning of novel classes. **For dynamic federated training, the goal extends beyond learning novel classes to effectively mitigate the forgetting of known classes. FOL, which only considers single-phase training, is not applicable in continual dynamic federated learning.**
>
> Secondly, **the training settings differ**. For FOL, given its focus on static federated training, clients' local data is also static, meaning there is no continual influx of streaming data, and **labeled data can be statically retained for training**. Furthermore, static federated training does not allow the participation of new clients. In contrast, new clients in dynamic federated training often have no data at all. Although their streaming data continues to arrive, it also lacks any labels. FedNovel specifically addresses the challenges posed by the continual arrival of streaming data. It assumes that during the novel class learning stage, **there will be no labeled data available, whether for known or novel classes**.
>
> Certainly, we understand your concern. We have already discussed the similarities and differences between FedNovel and FOL in the related work section and **removed the claim of the first work in our revised manuscript**. Furthermore, we have **modified the title of our paper to "Federated Continual Novel Class Learning"** to avoid any confusion and misunderstanding.
>
> >2. Insufficient comparison and discussion.
>
> Commonly used semi-supervised learning (SSL) typically assumes that the training data is partially labeled and that **both labeled and unlabeled data belong to the same label space**. The most prevalent method in SSL is progressive learning. This approach initially trains the model on labeled data, then tests it on unlabeled data, selecting high-confidence data to add back into the training set. As training goes on, an increasing amount of unlabeled data is incorporated, gradually improving the model's performance. In contrast, **the novel class learning problem that FedNovel focuses on has no labeled data available, and the novel classes do not overlap with the known classes**. Under such settings, even if a model trained on known classes is tested on novel classes, the confidence becomes meaningless due to the completely different label spaces.
>
> We understand your concerns. During the rebuttal phase, we conducted comparison experiments between our GAL approach and a federated semi-supervised learning work, SemiFL [F]. The experiment results are provided below, and we can observe that GAL can still substantially outperform SemiFL.
>
> | Dataset |  | CIFAR-100 |  |  |  | Tiny-ImageNet |  |  |  | ImageNet-Subset |  |
> |---|:---:|:---:|:---:|:---:|:---:|:---:|:---:|:---:|:---:|:---:|:---:|
> |  | known | novel | all |  | known | novel | all |  | known | novel | all |
> | SemiFL | 42.3 | 28.8 | 39.6 |  | 33.2 | 25.5 | 32.5 |  | 37.1  | 27.5 | 35.2 |
> | Our GAL | 72.6 | 45.8 | 67.3 |  | 57.6 | 32.7 | 55.0 |  | 55.8 | 29.7 | 50.5 |

---

> > ### Author Response · Authors · 2023-11-20
> > **Response to Reviewer RSBY Part [2]**
> >
> > >3. Limited novelty.
> >
> > Regarding your first question, **Prototype Contrastive Learning (PCL) is utilized in the known class learning stage** of our GAL, solely to enhance the model's capability for representation extraction. Moreover, **we have modified the standard PCL** by incorporating sample-wise comparison in the negative pair comparison. This strengthens the model's inter-class representation extraction ability. The **low-confidence sample rejection strategy is applied only in the data filtering of GAL** for both known and novel classes, and this filtering is only used before the initiation of novel class learning. In summary, the application of PCL and the low-confidence sample rejection strategy are merely preparatory steps before GAL initiates novel class learning. We need to mention that **our primary contributions lie in the global novel class number estimation and discriminative novel class learning** that occur after the initiation of novel class learning, corresponding to the Potential Prototype Merge (PPM) and Semantic Weighted Loss (SWL) in our GAL framework. As for why Opencon [C] may underperform compared to GAL, a plausible explanation is that although local contrastive learning enables local models to learn distinguishable clustering representations, in FedNovel, the novel label spaces differ across clients. This leads to various levels of shifts and rotations in the clustering representation distribution learned by local models, resulting in significant location differences of the same novel classes in the representation space.
> >
> > As for your second question, first, for the FedNovel problem, our proposed **GAL is not based on prototypical networks or prototype learning.** In GAL, to protect privacy, prototype communication is conducted only once, utilizing PPM to construct global prototypes. Once the global prototypes are constructed, they are used to initialize the novel class classifier, which is then updated with local training. After this, GAL does not include any prototype communication. The two works [D, E] you mentioned involve prototype sharing and communication in every global training round. In this setting, **GAL is more secure**. Furthermore, regarding the construction of prototypes, works D and E can reliably use labeled data, and as the training progresses, the prototypes constructed by them become increasingly reliable. However, **for GAL, there are no labeled data available to guide the construction of prototypes**, thus we use unsupervised clustering. Consequently, the prototypes constructed by GAL are highly unreliable. Therefore, **GAL treats the use of prototypes as a weakly supervised learning problem**, leading to the design of SWL. SWL innovatively optimizes the Euclidean distance between samples and prototypes, rather than cosine similarity, because the prototypes in GAL are constructed using k-means based on Euclidean distance. In addition, SWL does not arbitrarily consider only a single prototype but optimizes the relationships of a sample with all prototypes, effectively mitigating the impact of the unreliability of prototypes. In contrast, works D and E can directly utilize reliable and accurate prototypes for single pseudo labeling learning. From the problem setting to the specific solutions, GAL differs significantly from works D and E, and every aspect of GAL's design is logical and cohesively integrated.
> >
> > >4. Comparison with works A, B.
> >
> > Thank you for your suggestion. We have already provided a detailed discussion of the differences between our work and works A and B in the response to Question 1, and we have also included these discussions in our revised manuscript. Of course, we have conducted comparison experiments between works A, B and GAL within the FedNovel setting, with detailed results as follows. From the experimental results, it is evident that our proposed GAL still significantly outperforms works A and B.
> >
> > | Dataset |  | CIFAR-100 |  |  |  | Tiny-ImageNet |  |  |  | ImageNet-Subset |  |
> > |---|:---:|:---:|:---:|:---:|:---:|:---:|:---:|:---:|:---:|:---:|:---:|
> > |  | known | novel | all |  | known | novel | all |  | known | novel | all |
> > | FedoSSL [A] | 62.1 | 16.6 | 53.0 |  | 53.0 | 8.1 | 48.5 |  | 55.0  | 17.7 | 47.5 |
> > | AGCL [B] | 63.6 | 35.2 | 57.9 |  | 40.2 | 24.4 | 38.6 |  | 46.7 | 20.0 | 41.4 |
> > | Our GAL | 72.6 | 45.8 | 67.3 |  | 57.6 | 32.7 | 55.0 |  | 55.8 | 29.7 | 50.5 |

---

> > > ### Author Response · Authors · 2023-11-20
> > > **Response to Reviewer RSBY Part [3]**
> > >
> > > >5. Lack of theoretical proof for PPM.
> > >
> > > a) Firstly, we need to clarify that, to our best knowledge, **there currently isn't a rigorous framework available for the theoretical analysis of novel class number estimation** (NCNE) in fields of novel class discovery, openworld semi-supervised learning, and deep clustering. Now, let us attempt to analyze why existing NCNE methods perform poorly in FedNovel. Existing NCNE approaches are usually based on the assumption that if labeled known class data and unlabeled novel class data are mixed and clustered using an unsupervised clustering algorithm, when the labeled known class data is effectively separated, the unlabeled novel class data will also be effectively separated. **This assumption is problematic because for it to hold, two preconditions must be met: first, in the representation space, the distributions of known classes must be similar to those of novel classes; and second, the unsupervised clustering algorithm used must be capable of handling different data distributions.** Regarding the first precondition, **work [G]'s Theorem 1** states that when a learning model is trained using the EM (Expectation-Maximization) algorithm based on limited observation of training data distribution, it will map different labels to distinct representational distributions. In FedNovel, NCNE occurs before novel class learning. Therefore, only known classes undergo EM algorithm training based on limited observations, while novel classes do not undergo such training. **Therefore, there is a significant difference in the representation distribution of known and novel classes.** Regarding the second precondition, existing methods typically employ the k-means algorithm for unsupervised clustering. However, **k-means struggles with clustering tasks where cluster distributions vary significantly [H]**. There are also other influencing factors, such as that the representation augmentation performed on known class prototypes cannot accurately generate representations that conform to the original distribution statistics. Additionally, **local prototypes are a further abstraction on limited observation of novel class data distribution** (the last challenge mentioned on page 5 of the main paper when conducting NCNE in FedNovel).
> > >
> > > b) In contrast, our designed PPM can effectively work without the concern of the two aforementioned preconditions. Firstly, **PPM does not use known class data**, thus the difference in representational distributions between known and novel classes does not affect PPM's effectiveness. Moreover, upon careful comparison, we found that PPM shares the same design principle with incremental DBSCAN [I]. According to **Corollary 1 from work [I]**, as the parameter $\epsilon$ of DBSCAN increases, **PPM can identify and separate clusters of different densities.** Therefore, PPM can effectively locate high-density regions in FedNovel, leading to accurate NCNE.
> > >
> > > c) In addition to the experiments presented in the main text, we conducted **extensive experiments to validate the effectiveness of PPM**, including various scenarios of different novel class numbers across six datasets. For detailed experimental results, please refer to section B.5 in the appendix. According to these results, **PPM consistently provides the most accurate estimations,** with an estimation error significantly smaller than other methods. Certainly, in future work, we plan to explore more rigorous theoretical support for the effectiveness of GAL.

---

> > > > ### Author Response · Authors · 2023-11-20
> > > > **Response to Reviewer RSBY Part [4]**
> > > >
> > > > >6. The reasonableness of the experiment setting still needs to be considered.
> > > >
> > > > We chose to use the Dirichlet distribution to set the non-IID heterogeneity of FL because it aligns with the real-world FL scenarios and is currently the most commonly used, most popular, and most challenging non-IID distribution in state-of-the-art FL research. As you mentioned, applying the Dirichlet distribution in FedNovel does lead to data imbalance and variation in the degree of label overlap. Regarding your questions, after careful review, our response is as follows, assuming our understanding is correct. Of course, if we have misunderstood, please do inform us further.
> > > >
> > > > (1) Regarding your first question, we need to clarify that the Dirichlet distribution will cause imbalances in the labeled data during the known class learning stage, as well as in the unlabeled novel data during the novel class learning stage. However, it's important to mention that **there will be no labeled data during the novel class learning stage, from either known or novel classes**. **Therefore, the coexisting of imbalanced labeled and unlabeled data you mentioned is impossible**. Indeed, during the novel class learning, each client's novel class data distribution is different. Thus, the FedNovel solution needs to effectively address varying data distributions. Our GAL leverages PPM to construct global prototypes, providing a uniform standard for each client's local training. This enables GAL to effectively handle diverse data distributions.
> > > >
> > > > (2) Expanding the data imbalance caused by the Dirichlet distribution to extreme cases results in a scenario where different clients share some labels while having others that do not overlap. Therefore, **FedNovel deals with a hybrid situation where both these scenarios coexist, representing a more challenging problem**. In such a mixed scenario, GAL's PPM can effectively work. For example, for labels shared among clients, their corresponding local prototypes will be located in close proximity, forming high-density areas. PPM can accurately identify these areas and merge the prototypes contained within each area. As for labels that do not overlap between clients, PPM will segregate the corresponding local prototypes into separate areas. Thus, PPM can construct effective global prototypes in such mixed scenarios, allowing the subsequent novel class discriminative learning to naturally handle this case as well.
> > > >
> > > > (3) In our appendix, we have included additional experiments, which are relevant to data distribution, including a) experiments in the centralized setting (Section B.1); b) experiments with much more heterogeneous non-IID settings (Section B.2); c) experiments in settings with different data partitions (Section B.3). Please refer to these experimental results, as they may be helpful in addressing your concerns.
> > > >
> > > > [A] Towards Unbiased Training in Federated Open-world Semi-supervised Learning. ICML, 2023. [B] Federated Generalized Category Discovery, Arxiv, 2023. [C] Opencon: Open-world contrastive learning. TMLR, 2022. [D] Fedproto: Federated prototype learning across heterogeneous clients, AAAI, 2022. [E] Federated Semi-Supervised Learning with Prototypical Networks, Arxiv,2022. [F] Semifl: Semi-supervised federated learning for unlabeled clients with alternate training. NeurIPS, 2022. [G] Discover and Align Taxonomic Context Priors for Open-world Semi-Supervised Learning. NeurIPS, 2023. [H] Stability of K-Means Clustering. NIPS, 2006. [I] DBSCAN-like clustering method for various data densities. Pattern Analysis and Applications, 2020.

---

> > ### Comment · Reviewer_RSBY · 2023-11-23
> >
> > Thanks for the authors' response and detailed explanation. As pointed out by Reviewer a8Cy, the authors claim that FedNovel belongs to continual learning. However, this might result in a simpler and impractical setting for the following reasons: 1) Only engaging in one-step incremental new category discovery fails to demonstrate the ability for multi-step continuous learning, as seen in DM or iGCD[G]; 2) In comparison to fedossl[A] and Fed-GCD[B], the proposed FedNovel trains the model in two separate steps, using fully labeled and fully unlabeled data respectively, instead of training on mixed labeled and unlabeled data. Consequently, FedNovel contributes to a somewhat simpler and less realistic task.
> >
> > [A]: "Towards Unbiased Training in Federated Open-world Semi-supervised Learning," ICML, 2023. [B]: "Federated Generalized Category Discovery," Arxiv, 2023. [G]: "Incremental Generalized Category Discovery," ICCV, 2023. [H]: "Grow and Merge: A Unified Framework for Continuous Categories Discovery," NIPS, 2022.

---

> > > ### Author Response · Authors · 2023-11-23
> > > **Thanks for your reply! This is our further response, part [1].**
> > >
> > > Thank you for your reply. Regarding the new concerns you raised, **we do not believe that federated continual novel class learning is simpler or less realistic compared to static federated openworld learning. On the contrary, we consider addressing the former to be more meaningful and challenging than solving the latter.** In the real world, FL systems are always facing changing data distributions, such as the continuous emergence of new trends, news, products, and other information. **Static FL cannot cope with such new influxes of information, and the training costs for FL systems are substantial. It is impractical to train a new model for every new function.** Therefore, we need to enable FL systems to perform continual learning, which involves introducing new functionalities while maintaining the normal operation of existing ones. To achieve this goal, there are **several challenges:** (1) The streaming data for new functionalities arrives rapidly, cannot be labeled in a timely manner, and there may not be sufficient expertise to label it, resulting in mostly unlabeled incoming data. 2. The streaming data for new functionalities quickly consumes limited storage resources, including those for old functionalities, and data from old functionalities often cannot be stored for long due to copyright and privacy concerns. 3. New functionalities often come with new participants who usually do not have any old functionality data. These challenges directly correspond to the continuous arrival of unlabeled novel class data and the absence of known class data in FedNovel.
> > >
> > > Regarding the first reason you mentioned, the GAL we proposed for the FedNovel problem **is designed for multiple novel class learning stages.** After the known class learning stage, GAL continuously learns from the novel class data arriving at different times. **With each novel class learning stage, the FL model becomes capable of classifying accumulated novel classes.** Moreover, we have conducted experiments across multiple continual novel learning stages. We carried out multi-stage novel class learning on different datasets following GM [1] and iNCD [2], in particular, with each stage featuring a different number of novel classes. **The detailed experimental results are shown in Table 2 of our paper, where we observe that after several stages of novel class learning, GAL's advantages become even more pronounced**, significantly outperforming GM and iNCD which also allow for multi-stage novel class learning.
> > >
> > > Regarding the second reason you mentioned, we must reiterate that the learning difficulty of a learning task depends on the training setup, not on application scenarios. **To our best knowledge, we have not come across any assertions that continual learning is easier than openworld learning. Moreover, we are dealing with continual novel class learning, which means that we are not only facing challenges from novel class learning but also from continual learning.** As for continual novel class learning, **both GM (in the abstract) and iNCD (from lines 4 to 10 and lines 16 to 21 in the second paragraph of the introduction) indicate that its difficulty far exceeds that of regular novel class discovery (openworld learning)**. This is because, unlike conventional novel class discovery, continual novel class learning cannot use labeled known data. **This means that fewer resources are available for the same learning objectives, so why would the learning difficulty decrease? Therefore, we do not agree with the notion that the continual novel class learning scenario considered in FedNovel is simpler than static federated openworld learning.**

---

> ### Author Response · Authors · 2023-11-21
> **A Gentle Reminder of Further Feedback**
>
> Dear Reviewer RSBY,
>
> The conclusion of the discussion period is closing, and we eagerly await your response. We greatly appreciate your time and effort in reviewing this paper and helping us improve it.
>
> Thank you again for the detailed and constructive reviews. We hope our response is able to address your comments related to the comparison and discussion with more literature, novelty clarification, theoretical explanation of PPM, and issues about experiments. We take this as a great opportunity to improve our work and shall be grateful for any additional feedback you could give us.
>
> Best Regards,
>
> Authors of Paper 5479

---

> ### Author Response · Authors · 2023-11-22
> **Discussion Phase Ends in Less Than 24 Hours. We Eagerly Await Reviewer RSBY's Feedback.**
>
> Dear Reviewer RSBY,
>
> As the rebuttal discussion phase ends in less than 24 hours, we want to express our gratitude for your engagement thus far. We shall kindly remind you that after the 22nd, we are not allowed to respond to your further questions you may have. We really want to check with you whether our response addresses your concerns during the author-reviewer discussion phase. We have diligently addressed every concern and question you raised during the initial review, and our extensive efforts are aimed at enhancing the clarity and quality of our work.
>
> Your feedback is really important to us. We eagerly await any potential updates to your ratings, as they play a critical role in the assessment of our paper. We genuinely hope our responses have resolved your concerns and provided satisfactory explanations. Your thoughtful evaluation greatly aids in our paper's refinement and strength. We sincerely appreciate your dedication and time again.
>
> Best regards,
>
> Authors of Paper 5479

---

> ### Author Response · Authors · 2023-11-23
> **Final Reminder before Reviewer-Author Discussion Phase Closure for Reviewer RSBY**
>
> Dear Reviewer RSBY,
>
> Thank you again for the initial comments. As the Reviewer-Author Discussion phase is closing in less than 12 hours (Nov. 22nd AoE), we would greatly appreciate any feedback on our rebuttal. We fully understand that you may be busy at this time, but hope that you could kindly have a quick look at our responses and assess whether they have addressed your concerns and warrant an update to the rating. We would also welcome any additional feedback and questions.
>
> Best Regards,
>
> Authors of Paper 5479

---

> ### Author Response · Authors · 2023-11-23
> **Thanks for your reply! This is our further response, part [2].**
>
> In our response to Reviewer a8Cy, we elaborated on the differences in learning difficulty between label available novel class discovery and continual novel class learning. For label available NCDL, there are generally two approaches. The first one is based on semi-supervised learning (SSL), where the model is trained using labeled known class data while simultaneously selecting high-confidence unlabeled novel class data to participate in training. As training progresses, more novel class data are incorporated into training, enhancing the model's ability to extract representations for novel classes. The model's ability to extract representations for known classes is directly determined by the labeled known class data, **which means that the learning difficulty for known classes is fixed as long as the supervised training is applied.** Under such a scenario, if learning for known and novel classes is regarded as two separate tasks, **the SSL-based NCDL works like multi-task learning, leading to the learning of more task-shared features. Such task-shared features greatly benefit novel class learning.** The second approach to labeled available NCDL typically involves sufficient supervised pre-training using labeled known class data, followed by learning from unlabeled novel class data under the guidance of the labeled known class data. **With this approach, the model, trained on labeled known class data, tends to learn task-private features that are less beneficial to novel class learning compared to the task-shared features.** However, the second approach **still allows for the use of labeled known class data during novel class learning, facilitating cross-task transfer learning.** If there is any forgetting of the knowledge about known classes during novel class learning, it can be alleviated by retraining on the labeled known class data. Our proposed GAL is similar in training setup to the second method, but **differs in that there is no labeled known class data available to maintain the learned knowledge** during novel class learning. Although a significant portion of the knowledge the model has acquired about known classes is task-private, **there still exists some task-shared general knowledge. Without a dedicated design to combat knowledge forgetting, both task-shared and task-private knowledge will gradually fade away.** Once the task-shared knowledge is forgotten to a certain extent, novel class learning becomes an impossible task. Indeed, the difficulty of training for a task largely depends on the supervision information. **For labeled known class data, whether trained simultaneously with unlabeled novel class data or in supervised pre-training before novel class learning, the learning difficulty remains the same. For unlabeled novel class data, which lacks labels for novel classes, the learning difficulty is significantly higher, especially without even any supervision information from known classes.**
>
> **We have removed the claim of being the first work in our revised manuscript, but we still hope to inspire further exploration into the scenario of federated continual novel class learning through our work. It is by addressing these challenges that federated learning can be sustainably applied in the ever-changing real world.**
>
>
> [1] Grow and Merge: A Unified Framework for Continuous Categories Discovery. NIPS, 2022.
>
> [2] Class-incremental Novel Class Discovery. ECCV, 2022

---

### Official Review · Reviewer_a8Cy · 2023-10-31

**Soundness:** 3 good
**Presentation:** 3 good
**Contribution:** 3 good
**Rating:** 6
**Confidence:** 4

**Summary:**

This paper considers a novel FL scenario, where the data distribution involves dynamic and continual changes. Instead of naively integrating FL and conventional novel class discovery methods,  the authors propose a Global Alignment Learning (GAL) framework to estimate the number of novel classes and optimize the local training process in a semantic similarity-empowered reweighting manner. Extensive experiments have been conducted to demonstrate the efficiency of the proposed method.

**Strengths:**

[+] The whole paper is easy to understand, and well-written.

[+] The problem statement is very nice and clean. It also has some applications in practice.

[+] There is enough empirical evidence to support the main claims of the paper.

**Weaknesses:**

[-] It seems that open-world semi-supervised learning [1][2] also considers classifying both seen and unseen classes during the testing phase. Please compare it in related work.

[1] Open-world semisupervised learning. In International Conference on Learning Representations, 2022.

[2] Robust semi-supervised learning when not all classes have labels. In Advances in Neural Information Processing Systems, 2022.

[-] In section 4, the baselines on federated self-supervised learning methods are insufficient. It is suggested to add more FedSSL methods in the experimental part.

[3] Semifl: Semi-supervised federated learning for unlabeled clients with alternate training. Advances in Neural Information Processing Systems, 35:17871–17884, 2022.

**Questions:**

It is unclear why the value of $n_{size}$ is set as 2. A more detailed explanation should be added.

---

> ### Author Response · Authors · 2023-11-20
> **Response to Reviewer a8Cy**
>
> We would like to thank you for your positive and constructive comments on our work. We will address your concerns one by one as follows.
>
> >1. Discuss more related works.
>
> Thank you for your suggestion, we have discussed open-world semi-supervised learning (OSSL) in the Related Work of the revised paper. Although the setup of OSSL does share certain similarities with ours, there are still key differences: (1) **The data distribution differs**. In OSSL, the training data is centralized to locate a machine during training. However, in FedNovel, the data is distributed to multiple clients following the non-IID setting, which means the local data distributions of clients are distinct from each other. (2) **The data availability differs**. The data in OSSL are static, which means all labeled and unlabeled data can be accessed at the same time, thus they can be learned within a single training phase. In contrast, the data in FedNovel are dynamic. FedNovel first requires the model to learn only on labeled known classes, then moves to learn on unlabeled novel classes without access to labeled known class data. Therefore, in the novel class learning stage, there is one more challenge in FedNovel than OSSL, which is to preserve good performance on known classes.
>
>
> >2. Experimental results of SemiFL.
>
> Thanks for your suggestion. Similar to what we mentioned before, the key difference between SemiFL and FedNovel is also the **data availability**. The data in SemiFL are static while the data in FedNovel are dynamic. SemiFL can train on all labeled and unlabeled data in the same training phase, while FedNovel can only train on labeled classes in the first training phase and unlabeled classes in the following training phase. To further evaluate the effectiveness of our GAL, we also carry out the comparison experiments with SemiFL as below. Based on the results, GAL still achieves better performance than SemiFL.
>
> | Dataset |  | CIFAR-100 |  |  |  | Tiny-ImageNet |  |  |  | ImageNet-Subset |  |
> |---|:---:|:---:|:---:|:---:|:---:|:---:|:---:|:---:|:---:|:---:|:---:|
> |  | known | novel | all |  | known | novel | all |  | known | novel | all |
> | SemiFL | 42.3 | 28.8 | 39.6 |  | 33.2 | 25.5 | 32.5 |  | 37.1  | 27.5 | 35.2 |
> | Our GAL | 72.6 | 45.8 | 67.3 |  | 57.6 | 32.7 | 55.0 |  | 55.8 | 29.7 | 50.5 |
>
> >3. Why $n_{size}$ is set as 2.
>
> In our Potential Prototype Merging (PPM), $n_{size}$ is used to control the data scale of high-density regions. As the data in FedNovel are non-IID distributed among clients, **a particular client may have only a few data samples for certain novel classes. Besides, for a particular novel class, it may be possible that only a few clients own this class**. Setting a small $n_{size}$ can help our PPM not ignore the novel classes that correspond to a small number of local prototypes.

---

> > ### Comment · Reviewer_a8Cy · 2023-11-21
> >
> > I really appreciate the authors' response. Most of my concerns are well addressed. However, according to **Response to Reviewer RSBY**, the authors claimed that the application scenario belongs to continual learning, I'm afraid this setting would be easier than traditional static federated training, where the known and novel classes are simultaneously learned during the training phase.

---

> > > ### Author Response · Authors · 2023-11-21
> > > **Response to Reviewer a8Cy for Further Questions**
> > >
> > > Thanks very much for your further reply and questions. It's important to clarify that **the difficulty of novel class learning depends on the training setup, not the application scenario.** In static federated training, labeled known class data and unlabeled novel class data coexist. This situation belongs to the label available Novel Class Discovery and Learning (NCDL) we mentioned in our paper. For label available NCDL, there are generally two approaches. **The first one is based on semi-supervised learning (SSL)**, where the model is trained using labeled known class data while simultaneously selecting high-confidence unlabeled novel class data to participate in training. As training progresses, more novel class data are incorporated into training, enhancing the model's ability to extract representations for novel classes. The model's ability to extract representations for known classes is directly determined by the labeled known class data, which means that the **learning difficulty for known classes is fixed as long as the supervised training is applied.** Under such a scenario, if learning for known and novel classes is regarded as two separate tasks, the **SSL-based NCDL works like multi-task learning, leading to the learning of more task-shared features.** Such task-shared features greatly benefit novel class learning. **The second approach to labeled available NCDL typically involves sufficient supervised pre-training** using labeled known class data, followed by learning from unlabeled novel class data under the guidance of the labeled known class data. With this approach, the model, trained on labeled known class data, **tends to learn task-private features that are less beneficial to novel class learning compared to the task-shared features.** However, the second approach still **allows for the use of labeled known class data** during novel class learning, facilitating cross-task transfer learning. If there is any forgetting of the knowledge about known classes during novel class learning, it can be alleviated by retraining on the labeled known class data. Our proposed GAL is similar in training setup to the second method, but differs in that there is no labeled known class data available to maintain the learned knowledge during novel class learning. Although a significant portion of the knowledge the model has acquired about known classes is task-private, there still exists some task-shared general knowledge. Without a dedicated design to combat knowledge forgetting, both task-shared and task-private knowledge will gradually fade away. Once the **task-shared knowledge is forgotten to a certain extent, novel class learning becomes an impossible task**. **Indeed, the difficulty of training for a task largely depends on the supervision information. For labeled known class data, whether trained simultaneously with unlabeled novel class data or in supervised pre-training before novel class learning, the learning difficulty remains the same. For unlabeled novel class data, which lacks labels for novel classes, the learning difficulty is significantly higher, especially without even any supervision information from known classes.**

---

> ### Author Response · Authors · 2023-11-22
> **Do We Address Your Further Question? Further Check with Reviewer a8Cy**
>
> Dear Reviewer a8Cy,
>
> Thank you for your initial positive feedback and insightful suggestions. We appreciate your recognition of our efforts to address your concerns. We would like to check with you whether our response addresses your further concern about the learning difficulty of different application scenarios. We really hope our response can address your concern, certainly, we will be glad to answer any further questions.
>
> We value your input and sincerely hope you consider raising your rating based on the improvements we’re implementing. Your endorsement would greatly enhance the credibility of our work.
>
> Thank you once again for your time and valuable feedback.
>
> Best Regards,
>
> Authors of Paper 5479

---

> > ### Author Response · Authors · 2023-11-23
> > **Final Check with Reviewer a8Cy**
> >
> > Dear Reviewer a8Cy,
> >
> > Thank you again for the initial positive comments and further questions. As the Reviewer-Author Discussion phase is closing in less than 12 hours (Nov. 22nd AoE), we would check with you whether our new response has addressed your further questions. If possible, could you warrant an update to the rating? Your thoughtful evaluation greatly aids in our paper's refinement and quality. We sincerely appreciate your dedication and time again. We would also welcome any additional feedback and questions.
> >
> > Best Regards,
> >
> > Authors of Paper 5479

---

> > > ### Comment · Reviewer_a8Cy · 2023-11-23
> > >
> > > Thanks for the authors' response.  In my opinion, label-available NCDL methods [a] [b] [c] focus on a more practical scenario, where novel classes may appear in the unlabeled data. The proposed methods alleviate the training bias among labeled and unlabeled data caused by the class distribution mismatch. For novel class learning, they should avoid interference in the mixed training process since novel classes may be mistakenly classified into known classes. If the one-step training becomes two-step training, the above challenge may no longer exist.  The main challenge turns to how to avoid catastrophic forgetting and how to ensure consistency of the generated pseudo-labels between different clients. Based on this, I prefer to maintain my rating.
> > >
> > > [a] Open-world semisupervised learning. ICLR, 2022.
> > >
> > > [b] Robust semi-supervised learning when not all classes have labels. NeurIPS, 2022.
> > >
> > > [c] Towards Unbiased Training in Federated Open-world Semi-supervised Learning. ICML, 2023.

---

### Official Review · Reviewer_skcH · 2023-11-02

**Soundness:** 3 good
**Presentation:** 2 fair
**Contribution:** 3 good
**Rating:** 8
**Confidence:** 4

**Summary:**

The paper proposes a novel method to learn novel classes in federated learning with emerging unknown classes.

**Strengths:**

1. (Originality) The proposed method is novel by known-class representation learning and adaptive class merging without access to clients' data.
2. (Clarity) The paper is clear in techniques. Methods are well formulated and motivated. Sufficient details are provided for the experiments.
3. (Significance) The proposed method is sufficiently evaluated in multiple datasets, models including small-scale sets (like Cifar10, or Cifar100) and large-scale sets (ImageNet). In all of these experiments, the proposed methods outperform the baselines in both known class and novel class evaluations.
4. (Quality) Extensive experiments evaluate the method in multiple dimensions. Importantly, multiple federated learning is demonstrated to be integrable with the proposed method.

**Weaknesses:**

1. The authors claim their contribution as a federated novel-class learning without compromising privacy. However, it is unclear how the existing federated novel-class learning methods compromise privacy. Importantly, the definition of private information is vague. It seems that the number of novel classes is thought to be private, which however is not necessarily true. Without specification on the privacy definition, there also lacks sufficient justification for how the proposed method will protect privacy. Though I appreciate the empirical results of privacy evaluation, the authors should clarify the meaning of privacy and adjust the claim of privacy.

**Questions:**

* What is the definition of private information in the paper?

---

> ### Author Response · Authors · 2023-11-20
> **Response to Reviewer skcH**
>
> First of all, we really apprepiate your positive and constructive comments on our work. We will address your concerns one by one.
>
> >1. The number of novel classes
>
> **In FedNovel, the number of novel classes is not private information.** This can be verified by works [1] in federated open-world semi-supervised learning, where the server is assumed to know the exact number of unseen classes before training. In contrast, the proposed GAL can estimate the number of novel classes by using our designed Potential Prototype Merging (PPM).
>
> [1] Towards Unbiased Training in Federated Open-world Semi-supervised Learning. ICML, 2023.
>
> >2. Definition of private information
>
> **The private information that needs to be protected in FedNovel is the meaningful semantic information of raw data for every client.** If existing data reconstruction attacks can't recover such semantic information from the prototypes sent by each client, we can say that our GAL does not leak private information. We have carried out experiments of launching data reconstruction attacks on the prototypes in GAL. The detailed results are shown in Appendix B.8, and the reconstructed images shown in Figure 3 look very similar to random noise in human eyes. To further verify that semantic information is not contained in the reconstructed images, we compute the Structural Similarity (SSIM) between reconstructed images and all images from each class. Then we check whether there is a set of classes that always share higher similarity with the reconstructed images and whether this class set is major included in the clusters corresponding to the attacked prototypes. Specifically, we choose the top 50% classes (10 out of 20) according to the average SSIM and set their corresponding dimensions as 1 in a zero vector. We also create another zero vector with the major component classes of a prototype as 1. Then we calculate the Manhattan Distance (MD) between these two vectors. For a better comparison, we also obtain the same vectors for random noise images and calculate the MD between vectors of reconstructed and random noise images. **According to the following results, we can conclude that data reconstruction attacks cannot recover meaningful semantic information, thus our GAL is privacy-preserving.**
>
> |Manhattan Distance|CIFAR-100|Tiny-ImageNet|ImageNet-Subset|
> |:---:|:---:|:---:|:---:|
> |with the vectors of major cluster components|9.7|10.1|10.5|
> |with the vectors of gaussian noise images|10.1|9.9|10.2|

---

> > ### Comment · Reviewer_skcH · 2023-11-21
> > **Thank you for the rebuttal**
> >
> > Thanks for authors' responses, which addressed all of my concerns.

---

> > > ### Author Response · Authors · 2023-11-22
> > > **Thanks to Reviewer skcH**
> > >
> > > Thanks so much for checking our response. We are so glad that your concern could be addressed. We also appreciate any further feedback and questions. Thanks again for your initial comments and dedicated efforts.

---

### Author Response · Authors · 2023-11-20
**Revision Change Summary**

[Title] We have changed the paper title to "Federated Continual Novel Class Learning" to avoid misunderstandings.

[Figure 1] We have incorporated more information into Figure 1 to more clearly depict the overall proposed GAL framework for the FedNovel problem.

[Related Work] We have discussed Federated Openworld/Openset Learning and Openworld Semi-Supervised Learning in the section of Related Work, in particular, we have talked about the difference between them and FedNovel.

[Appendix Section C] We have included an overall optimization workflow of our proposed GAL framework from known class learning to novel class learning.

---

### Author Response · Authors · 2023-11-20
**Thanks for All Reviewers**

We would like to thank all the reviewers for their insightful comments and constructive suggestions. Below we provide our detailed response to each reviewer. We have uploaded a revised version of our submission, with major changes highlighted in magenta (there are also other minor changes in wording and typos). New experiments are also conducted, and the detailed results are reported in the response to you, respectively. Thank you and we look forward to any further feedback and discussion.

---

### Meta-Review · Area_Chair_36wm · 2023-12-01

**Metareview:**

This paper considers a federated learning setting in the scenario of novel class discovery. To solve this problem, the authors propose a Global Alignment Learning (GAL) framework to estimate the number of novel classes and optimize the local training process in a semantic similarity-empowered reweighting manner. Extensive experiments have been conducted to demonstrate the efficiency of the proposed method.

**Strengths**

- The whole paper is easy to understand, and well-written.

- The problem statement is clean. It also has some applications in practice.

- The proposed method is reasonable and achieves very good performance over compared methods.

- Extensive experiments are conducted to demonstrate the effectiveness of the proposed method.


**Weaknesses**

After rebuttal, most of the concerns raised by the reviewers are solved. However, several weaknesses still remains by two reviewers.

- The proposed setting indeed is not very new. There are two highly related works, [A, B], which study the federated open-world semi-supervised learning, or called federated generalized novel class discovery, were proposed before. However, this paper does not well acknowledge these two works in the manuscript, especially in the introduction. This somewhat may mislead the readers to ignore these two works. In addition, as agreed by Reviewer a8Cy and Reviewer RSBY, compared to the settings in [A, B], the proposed setting is somewhat simpler, while this concern is not solved during rebuttal.

- In addition, even with revising, the comparison with [A, B] in the paper is still unclear, including, what is the difference between the proposed setting and [A, B]; what is the new challenging in the proposed setting; lack of acknowledgement of the first federated open-world semi-supervised learning and novel class discovery works. Also, the comprehensive comparisons and analyses with other related settings (e.g., federated lifelong learning, unsupervised lifelong learning) are insufficient.

- The setting name should be carefully reconsidered. Initially, the authors name the setting as Federated Novel Class Learning while rename it as Federated Continual Novel Class Learning in the title of the revision. This is more accurate. However, the authors did not revise it for the remaining parts in the paper.  There is a significant difference between with and without [continual], so the authors should well claim this in the revision. In addition, why call it as novel class learning instead of the well-known [Novel Class Discovery, Generalized Category Discovery] is unclear.

- Minor: It is unexpected that the authors use medical imaging as examples in the framework but only conduct experiments on the generic classification datasets (CIFAR, ImageNet).

[A] Towards Unbiased Training in Federated Open-world Semi-supervised Learning. ICML, 2023.

[B] Federated Generalized Category Discovery, Arxiv, 2023

In conclusion, this paper is well-written and proposes a relatively new setting for novel class discovery. Extensive experiments are conducted to validate the proposed method among several settings. However, the comparison with previous highly related works [A, B] is insufficient and unclear. In addition, this paper lacks well-acknowledging the existing of [A, B] in a right statement (e.g., clear them in the introduction) . The AC thus thinks this paper should undergone a significant revision and is not ready to meet the requirement of ICLR at this point.

**Justification For Why Not Higher Score:**

The comparison with previous highly related works is insufficient and unclear. In addition, this paper lacks well-acknowledging the existing of [A, B] in a right statement (e.g., clear them in the introduction) .

**Justification For Why Not Lower Score:**

N/A

---

### Decision · Program_Chairs · 2024-01-16

Reject